

# Comparative systematics and phylogeography of *Quercus* Section *Cerris* in western Eurasia: inferences from plastid and nuclear DNA variation

Marco Cosimo Simeone[1], Simone Cardoni[1], Roberta Piredda[2],
Francesca Imperatori[1], Michael Avishai[3], Guido W. Grimm[4] and Thomas Denk[5]

[1] Department of Agricultural and Forestry Science (DAFNE), Università degli Studi della Tuscia, Viterbo, Italy
[2] Stazione Zoologica Anton Dohrn, Napoli, Italy
[3] Jerusalem Botanical Gardens, Hebrew University of Jerusalem, Jerusalem, Israel
[4] Orleans, France
[5] Department of Palaeobiology, Swedish Museum of Natural History, Stockholm, Sweden

Corresponding author
Marco Cosimo Simeone,
mcsimeone@unitus.it

## ABSTRACT

Oaks (*Quercus*) comprise more than 400 species worldwide and centres of diversity for most sections lie in the Americas and East/Southeast Asia. The only exception is the Eurasian sect. *Cerris* that comprises about 15 species, most of which are confined to western Eurasia. This section has not been comprehensively studied using molecular tools. Here, we assess species diversity and provide a first comprehensive taxonomic and phylogeographic scheme of western Eurasian members of sect. *Cerris* using plastid (*trnH-psbA*) and nuclear (5S-IGS) DNA variation with a dense intra-specific and geographic sampling. Chloroplast haplotypes primarily reflected phylogeographic patterns originating from interspecific cytoplasmic gene flow within sect. *Cerris* and its sister section *Ilex*. We identified two widespread and ancestral haplotypes, and locally restricted derived variants. Signatures shared with Mediterranean species of sect. *Ilex*, but not with the East Asian *Cerris* oaks, suggest that the western Eurasian lineage came into contact with *Ilex* only after the first (early Oligocene) members of sect. *Cerris* in Northeast Asia had begun to radiate and move westwards. Nuclear 5S-IGS diversification patterns were more useful for establishing a molecular-taxonomic framework and to reveal hybridization and reticulation. Four main evolutionary lineages were identified. The first lineage is comprised of *Q. libani*, *Q. trojana* and *Q. afares* and appears to be closest to the root of sect. *Cerris*. These taxa are morphologically most similar to the East Asian species of *Cerris*, and to both Oligocene and Miocene fossils of East Asia and Miocene fossils of western Eurasia. The second lineage is mainly composed of the widespread *Q. cerris* and the narrow endemic species *Q. castaneifolia*, *Q. look*, and *Q. euboica*. The third lineage comprises three Near East species (*Q. brantii*, *Q. ithaburensis* and *Q. macrolepis*), well adapted to continental climates with cold winters. The forth lineage appears to be the most derived and comprises *Q. suber* and *Q. crenata*. *Q. cerris* and *Q. trojana* displayed high levels of variation; *Q. macrolepis* and *Q. euboica,* previously treated as subspecies of *Q. ithaburensis* and *Q. trojana,* likely deserve independent species status. A trend towards inter-specific crosses was detected in several taxa; however, we found no clear evidence of a hybrid origin of *Q. afares* and *Q. crenata,* as currently assumed.

## INTRODUCTION

Studies on the genetic diversity of forest species across their distributional ranges are relevant for genetic resource inventories and devising conservation strategies (*Pautasso, 2009*). Comparative phylogeographic studies may further reveal complex spatial variation patterns within groups of closely related species (sibling lineages, species aggregates), shaped by partly antagonistic evolutionary and ecological processes. The detailed genetic information can be used to address taxonomic questions, assist biodiversity surveys, and implement species conservation and future landscape management strategies (*Barak et al., 2016*).

Oaks (*Quercus* L.) are an ideal model for comparative phylogeographic studies. They are common, often (co-)dominant vegetation elements and include several widely distributed and ecologically diverse species (*Camus, 1936–1954*; *Schwarz, 1936–1939*). Oaks also have a strong potential for ecological adaptation, accompanied by substantial leaf morphological variability and a high potential for introgression and reticulate evolution (e.g., *Burger, 1975*; *Van Valen, 1976*; *Petit et al., 2004*; *McVay, Hipp & Manos, 2017*). Therefore, regional estimates about the number of oak species have been strongly deviating (see e.g., http://www.ipni.org). At the same time, taxonomic ambiguity and reticulate evolution during formation of species make sampling and phylogenetic tree inference difficult.

*Quercus* has recently been formalized as two subgenera with eight sections (*Denk et al., 2017*). The predominantly Nearctic subgenus *Quercus* includes the sections *Lobatae* (Americas), *Protobalanus* (western North America), *Ponticae* (two disjunct species in southwestern Georgia/ northeastern Turkey and northern California/southwestern Oregon), *Virentes* (southeastern US into Mesoamerica), and *Quercus* (with most species in North America and about 30 species in Eurasia). The Palearctic-Indomalayan subgenus *Cerris* includes sections *Cyclobalanopsis* (East Asia), *Ilex* and *Cerris* (across Eurasia). For some of these groups, detailed infragroup phylogenies and assessment of main biogeographic patterns have recently been published (e.g., *Hipp et al., 2014*: North American sect. *Quercus*; *Cavender-Bares et al., 2015*: sect. *Virentes*; *Vitelli et al., 2017*: western Eurasian members of sect. *Ilex; Deng et al., 2018*: sect. *Cyclobalanopsis*). The understudied sect. *Cerris* (*Cerris* oaks) currently includes 13 species and a few unresolved taxa (Table 1) occurring from the Atlantic coasts of the Iberian Peninsula and Morocco to Japan. They thrive under a variety of climates (Köppen-Geiger climate types): cold steppe (*BSk*) and warm temperate or snow climates with different precipitation regimes, including more arid summer-dry and (more) mesic per-humid and winter-dry regimes (*Cs, Cf, Cw, Ds, Df, Dw*; *Kottek et al., 2006*; *Peel, Finlayson & McMahon, 2007*; *Rubel et al., 2016*). These oaks are deciduous or semi-evergreen (leaf lifespan up to 12–18 months) trees up to 30 m tall, characterized by pollen with scattered verrucate ornamentation, imbricate,

**Table 1** **Species and taxa included in *Quercus* Section *Cerris*.** Nomenclature followed *Govaerts & Frodin (1998)*; species investigated in the present study are bolded. Taxonomic remarks and species distributions according to \**Govaerts & Frodin (1998)* and \*\**Menitsky (2005)*.

| Taxon | Taxonomic remarks | Distribution |
| --- | --- | --- |
| **Q. afares Pomel** | | Endemic, Algerian and Tunisian Tell Atlas |
| **Q. brantii Lindl.** | | S and SE Anatolia to Iran, Lebanon |
| **Q. castaneifolia C.A. Mey** | | Endemic; SE Caspian Sea, Azerbaijan to Iran |
| **Q. cerris L.** | | E and C Mediterranean, Balkans |
| **Q. crenata Lam.** | \*Poorly known | Endemic, Italian peninsula |
| **Q. libani Oliv.** | | SE Anatolia to Iran |
| **Q. look Kotschy** | \*Synonym of *Q. ithaburensis* ssp. *ithaburensis*; \*\*hybrid *Q. ithaburensis* × *Q. libani* | Endemic, Lebanon to Anti-Lebanon mountain range |
| **Q. ithaburensis Decne.** | \*\*Including subsp. *macrolepis* (Kotschy), distributed in the European part of the range, and subsp. *ithaburensis* (Decaisne), in the Middle East | E Mediterranean, SE Italy to Palestine |
| **Q. trojana Webb** | \*Including subsp. *trojana* and subsp. *euboica* (Papaioann.) K.I.Chr., endemic of Euboea (Greece) | Anatolia, Aegean to SE Italy |
| **Q. suber L.** | | C and W Mediterranean |
| Q. acutissima Carruth. | | E and SE Asia |
| Q. chenii Nakai | | E Asia |
| Q. variabilis Blume | | E and SE Asia, Japan |

recurved and elongated cupule scales, tomentose endocarp, and pointed, elongated styles; their leaves are generally toothed or lobed and usually with a mucronate apex (*Denk et al., 2017*). Based on the fossil record and molecular differentiation patterns, it has been suggested that sect. *Cerris* evolved from sect. *Ilex*, possibly in Europe during the Miocene (*Denk & Grimm, 2010*; *Simeone et al., 2016*). However, this scenario needs to be revised as unambiguous fossils of sect. *Cerris* are now known from early Oligocene deposits of northeastern Russia (Russian Far East), and sect. *Ilex* appears to have been present in middle Eocene strata of southern China (*Denk et al., 2017*).

At present, sect. *Cerris* is most diverse in western Eurasia, with eight commonly accepted species (*Govaerts & Frodin, 1998*) and some additional forms (*Q. crenata, Q. look, Q. trojana* subsp. *euboica, Q. ithaburensis* subsp. *macrolepis*) of disputed or unresolved status. The ranges of *Cerris* oaks vary substantially in size and in the degree of contact with other species of the same section and the sister section *Ilex*. The Qinghai-Tibet Plateau and the Himalayan front-hills, home of several species of sect. *Ilex*, separate the East Asian and the western Eurasian taxa. The central and eastern Mediterranean region comprise most of the group's diversity (Anatolia and Levant to S. Italy and N.E. Algeria, eight species), which decreases westward (Iberian Peninsula and Morocco, one species) and eastward (Iran/ Iraq, three species; *Browicz & Zieliński, 1982*). Throughout the Mediterranean, the distribution ranges of *Cerris* oaks overlap with section *Ilex*, particularly *Q. ilex*. Two species have broad distributions (*Q. suber,* the 'Cork Oak', partly due to cultivation, and *Q. cerris*) and three are geographically extremely limited (*Q. afares*, *Q. castaneifolia*, *Q. look*). A hybrid origin has been postulated for *Q. crenata* (infra-sectional hybrid) and *Q. afares* (inter-sectional hybrid; *Mir et al., 2006*; *Conte, Cotti & Cristofolini, 2007*). Other occasional infra-sectional
hybrids have also been described as morphologically intermediates (see *Menitsky, 2005*). *Quercus suber* shows interfertility with the partly sympatric *Q. ilex* of sect. *Ilex* (*Burgarella et al., 2009*).

Detailed phylogeographic inferences are so far available only for two East Asian species of sect. *Cerris* thriving in temperate and subtropical broad-leaved forests in eastern Asia, *Q. acutissima* and *Q. variabilis*, based on plastid DNA sequence analyses (*Chen et al., 2012*; *Zhang et al., 2015*). In both cases, high genetic diversity but weak phylogeographic structure was found, explained with recent (Pleistocene) speciation and post-glacial re-expansion of lineages. In western Eurasia, the two most widespread species (*Q. suber* and *Q. cerris*) were studied using plastid microsatellite variation (*Magri et al., 2007*; *Bagnoli et al., 2016*). Geographically structured gene pools were detected and their formation attributed to the Oligocene (*Q. suber*) and Pleistocene (*Q. cerris*), respectively. Local investigations focussing on conservation were conducted on *Q. trojana* in Italy and *Q. libani* in Iran using nuclear microsatellites (*Khadivi-Khub et al., 2015*; *Carabeo et al., 2017*). Finally, species of the entire section were included in DNA barcoding projects and studies on molecular macroevolution (e.g., *Denk & Grimm, 2010*; *Simeone et al., 2013*; *Simeone et al., 2016*), but relied on a limited number of individuals.

At present, firm species delineation and phylogenetic inferences using sequence data are difficult in *Quercus*. All plastid data assembled so far showed strong disagreement with taxonomy and systematics (*Grímsson et al., 2016*; *Denk et al., 2017*) and nuclear regions with sufficient levels of variation, especially when closely related species are involved, are not yet available (*Muir, Fleming & Schlotterer, 2001*; *Oh & Manos, 2008*; *Hubert et al., 2014*). Unrepresentative inter- and (especially) intra-specific samplings constitute additional obstacles (*Manos, Zhou & Cannon, 2001*; *Bellarosa et al., 2005*; *Chen et al., 2017*), that partially reduce the potential for thorough within-lineage species delineation, even with high-resolution phylogenomic approaches (*Hipp et al., 2014*; *Hipp et al., 2018*; *Cavender-Bares et al., 2015*). Nevertheless, the efficacy of the nuclear ribosomal 5S rDNA intergenic spacer (in contrast to the internal spacers of the 35S rDNA, ITS1 and ITS2) to resolve species differentiation in western Eurasian members of Sect. *Cerris* was demonstrated by *Denk & Grimm (2010)*. Being potentially affected by incomplete lineage sorting, intra-array recombination and intragenomic competition, this marker requires a special analysis framework (cloning and *host-associate* analysis; cf. *Göker & Grimm, 2008*) but enables tracking of reticulate evolutionary signatures.

On the other hand, being largely controlled by provenance and decoupled from speciation, plastid data are important to trace the radiation of lineages in space and time (*Pham et al., 2017*). In this view, the evolutionary trajectories of sect. *Cerris* and *Ilex* share some traits that need to be fully addressed. Two recent studies on the Mediterranean members of sect. *Ilex* (*Simeone et al., 2016*; *Vitelli et al., 2017*) found three main plastid haplotype groups, with distinct geographic distribution and phylogenetic features: (1) 'Euro-Med', comprising the most distinct haplotypes dominating in the western Mediterranean, a plastid lineage that diverged before the radiation of plastid pools in Subgenus *Cerris*, (2) 'WAHEA' (i.e., West Asian–Himalayan–East Asian), distributed from Anatolia/Levant to East Asia, and sister to (3) 'Cerris-Ilex', centred on the Aegean Sea and

shared with co-occurring members of sect. *Cerris*. Haplotype variation of the *trnH-psbA* intergenic spacer was determinant for the phylogeographic inferences; this marker also showed the highest variation rate among several plastid regions in 35 Chinese oak species (*Yang et al., 2017*), and in the comprehensively studied *Q. acutissima* and *Q. variabilis* (*Zhang et al., 2015*; *Chen et al., 2012*).

Clearly, full comprehension of the drivers of speciation of sect. *Cerris* requires information from both genomes based on extensive geographic and taxonomic sampling. In this work, we investigated 5S-IGS and *trnH-psbA* molecular diversity in western Eurasian sect. *Cerris* using a comprehensive intra- and inter-specific sampling. Our objectives were: (i) to assess species coherence and delimitation, (ii) to infer inter-species relationships, (iii) to gain insight into the origin and diversification of the group. We further hypothesize that signals from both markers may combine to reveal potential zones of secondary genetic contact between already established species and to gain insight into the putative hybrid status of *Q. crenata* and Q. *afares*. All data will likely contribute to identify biogeographic hotspots for further in-depth studies on species diversity, isolation, introgression and natural hybridisation in oaks.

## MATERIALS AND METHODS
### Plant material and DNA sequencing
We combined previously studied (*Denk & Grimm, 2010*) and new material to develop a sampling design of 221 individuals completely covering the taxonomic breadth and distribution range of *Quercus* sect. *Cerris* in western Eurasia (File S1); some individuals with intermediary morphology were labelled as presumed hybrids. 158 individuals were studied for the first time, and new plastid data were generated for the previously and newly studied individuals. DNAs were extracted from silica gel-dried leaf samples with the DNeasy Plant minikit, following the manufacturer's instructions. The *trnH-psbA* intergenic spacer was amplified and sequenced following *Simeone et al. (2016)*. The nuclear ribosomal 5S intergenic spacer (5S-IGS) was amplified with the primer pair 5S14a and 5S15 (*Volkov et al., 2001*; *Denk & Grimm, 2010*). Individual PCR fragments were ligated into a pGEM-T easy vector (Promega). The ligation mixtures, purified with the Illustra GFX PCR DNA Purification kit (GE Healthcare), were used to transform *E. coli* strain XL1-Blue electroporation-competent cells (recA1, endA1, gyrA96, thi-1, hsdR17, supE44, relA1, lac, [F' proAB, lacIqZ ΔM15, Tn10 (tetr)]). The positive clones, selected on LB/Ampicillin plates, were identified by colony PCR using the amplification primers. Five to ten recombinant clones per individual were sequenced with the vector-specific universal primers (SP6/T7) at LGC Genomics (Augsburg, Germany). The GenBank *trnH-psbA* sequences of several East Asian members of sect. *Ilex* (*Q. baloot*, *Q. floribunda*, *Q. phylliraeoides*, *Q. semecarpifolia*, *Q. baroni*, *Q. dolicholepis*, *Q. spinosa*), East Asian species of sect. *Cerris* (*Q. acutissima, Q. variabilis*) and further sequence accessions of the investigated group were included in the analyses. In addition, all 5S IGS sequences of *Quercus* sect. *Cerris* available on GenBank were included in the final dataset, and the sequences of *Q. baloot* and *Q. floribunda* were used as outgroups, based on *Denk & Grimm (2010)*; all GenBank accession numbers are reported in File S1.

## Data analyses

Eye-checked electropherograms were aligned in MEGA7 (*Kumar, Stecher & Tamura, 2016*). Highly dissimilar clone sequences showing no BLAST match with the targeted regions (*Altschul et al., 1990*) were filtered. Final multiple alignments were obtained with ClustalW 1.81 (*Thompson, Higgins & Gibson, 1994*) and checked by eye. The diversity of the investigated regions was evaluated with MEGA7 and DnaSP5.1 (*Librado & Rozas, 2009*). Median-joining (MJ) haplotype networks for the *trnH-psbA* region were inferred with Network 4.6.1.1 (http://www.fluxus-engineering.com/), treating gaps as fifth state. The MJ algorithm was invoked with default parameters (equal weight of transversion/transition), in order to handle large datasets and multistate characters.

After removal of identical clones, the total 5S-IGS sequences were used to build a Maximum likelihood (ML) tree with RAxML v8.2 (*Stamatakis, 2014*) using the in-built GTR + $\Gamma$ model with the 'extended majority-rule consensus' criterion as bootstopping option (*Pattengale et al., 2009*). To infer inter-individual relationships, we applied the approach described by *Göker & Grimm (2008)* that allows transformation of data matrices of 'associates' (here: cloned sequences) into 'hosts' (here: individuals). The program g2cef (available at http://www.goeker.org/mg/distance/) was used to transform the primary character matrix ('associates', total cloned sequences) into a character consensus matrix of the individuals ('hosts') using an association file defining the list of clone sequences belonging to the same individual. The uncorrected pairwise distances of the primary character matrix ('associates', total cloned sequences) was calculated and used as input to the program pbc (*Göker & Grimm, 2008*). This program allows transforming the primary inter-clone pairwise distance matrix into inter-individual distances matrices using different flavours, of which the 'Phylogenetic Bray-Curtis' (PBC) transformation performed best in the original study that compared data sets with similar properties than our data set. Here, we applied three of the distance transformations tested by *Göker & Grimm (2008)*, in addition to PBC distances (option -b) also the minimum (MIN; -i) and average (AVG; -a) inter-individual clonal distances. AVG, MIN and PBC distance matrices were generated setting different minimum number of associates per host (-m option); $m = 4$ (the number of cloned sequences obtained in most individuals) was then used to infer a phylogenetic network using the Neighbour-Net (NN) algorithm (*Bryant & Moulton, 2004*) implemented in SplitsTree4 (*Huson & Bryant, 2006*).

## RESULTS

In total, 221 individuals effectively covering the taxonomic range of western Eurasian sect. *Cerris* were analysed (Table 1, File S1). Sequence quality was high for both marker regions and unambiguous electropherograms were obtained for about 90% of the investigated samples. The primary data matrixes comprised 207 plastid (*trnH-psbA*) and 856 nuclear (5S-IGS) sequence accessions. The nuclear data (192 individuals) included 651 newly sequenced clones and 205 accessions from *Denk & Grimm (2010)*. Ten *Q. baloot/Q. floribunda* sequences, used here as outgroups (cf. *Denk & Grimm, 2010*), extended the final dataset to 866 sequenced clones. Individual sequences recovered from positive 5S-IGS clones varied

**Table 2  Diversity values of the *trnH-psbA* IGS in the investigated dataset.**

| Dataset | N | L | p | H | h | Hid | S | PICs |
|---|---|---|---|---|---|---|---|---|
| West Eurasian species | 207 | 503 | 0.000–0.008 (±0.004) | 12 | 0.515 | H1–H12 | 6 (27) | 6 |
| Q. afares | 7 | 491 | 0.000 | 1 | 0.000 | H1 | 0 | 0 |
| Q. brantii | 7 | 487 | 0.000 | 2 | 0.476 | H5, H6 | 0 (1) | 0 |
| Q. cerris | 52 | 493 | 0.000–0.002 | 8 | 0.538 | H1–H3, H5–H7, H9, H10 | 1 (9) | 1 |
| Q. castaneifolia | 2 | 491 | 0.000 | 1 | 0.000 | H1 | 0 | 0 |
| Q. crenata | 6 | 491 | 0.000 | 1 | 0.000 | H1 | 0 | 0 |
| Q. trojana[a] | 45 | 493 | 0.000–0.002 | 3 | 0.369 | H1, H2, H4 | 1 (3) | 1 |
| Q. ithaburensis[b] | 33 | 493 | 0.000–0.002 | 5 | 0.655 | H1, H2, H4, H6, H9 | 1 (8) | 1 |
| Q. look | 3 | 488 | 0.000 | 2 | 0.667 | H9, H10 | 0 (1) | 0 |
| Q. suber | 47 | 501 | 0.000–0.006 | 3 | 0.303 | H1, H11, H12 | 3 (18) | 3 |
| Q. libani | 5 | 493 | 0.000–0.002 | 3 | 0.700 | H1, H3, H8 | 1 (8) | 0 |
| Q. acutissima[c] | 401 | 564 | 0.000–0.004 | 10 | n.d. | /[d] | 4 (79) | 0 |
| Q. variabilis[c] | 528 | 594 | 0.000–0.004 | 11 | n.d. | /[d] | 2 (99) | 0 |

**Notes.**

N, number of sequences; L, Aligned length (bp) with the inversion deleted; p, uncorrected p-distance range (STD); H, Number of identified haplotypes (gaps included); h, Haplotype diversity; Hid, haplotype code; S, Number of polymorphic sites (gaps included); PICs, Number of Parsimony Informative Characters.

[a]Including subsp. *euboica*.

[b]Including subsp. *macrolepis*.

[c]GenBank haplotype accessions: KT152191, KT152192, KT152193, KT152194, KT152195, KT152196, KT152197, KT152198, KT152199, KT152200, JF753573, JF753574, JF753575, JF753576, JF753577, JF753578, JF753579, JF753580, JF753581, JF753582, JF753583, KM210647, HE585136.

[d]No haplotype shared with the West Eurasian dataset, one haplotype shared between the two East Asian species.

from one (four samples) to 10, with most samples represented by four sequences (70 samples), followed by five and three sequences (44 and 35 samples, respectively). Multiple alignments of both marker regions were straigthforward. A 34-bp inversion occurring in the *trnH-psbA* region of 14 samples was replaced with its reverse-complementary sequence. Since it did not show further mutations it was deleted and a binary character was inserted to keep record of it.

## Plastid *trnH-psbA* diversity and biogeography

After removing the 34-bp inversion, the *trnH-psbA* marker showed pairwise uncorrected p-distances ranging between zero and 0.008 (Table 2). The highest intra-specific distance (0.006) was found in *Q. suber*; four species showed similar values (0.002; *Q. cerris*, *Q. ithaburensis*, *Q. trojana*, *Q. libani*), while the marker variation in the remaining taxa converged to zero.

The total matrix was 503-bp characters long, including several indels (1–8 bp) and six polymorphic sites resulting in twelve haplotypes (labelled H1–H12) with a medium overall diversity ($h = 0.515$). Haplotype H1 hit 100% sequence identity with three non-representative individuals assigned to East Asian species of sect. *Ilex* in Genbank (haplotype list, occurrence and gene bank matches shown as Files S1 and S2). It was the most common haplotype, occurring in 68.6% of individuals and all taxa except *Q. brantii*, *Q. look*, and *Q. ithaburensis* subsp. *ithaburensis* (henceforth *Q. ithaburensis*). Haplotypes H2, H5–H7 and H11 showed 100% sequence identity with Mediterranean members of sect. *Ilex* (*Simeone et al., 2016*; *Vitelli et al., 2017*). H2 is the second most frequent haplotype, found in 10.6% of

*Q. cerris*, *Q. trojana*, *Q. ithaburensis* subsp. *macrolepis* (henceforth *Q. macrolepis*) samples from Turkey, the Balkans and Italy; H5–H7 were found in *Q. brantii*, *Q. cerris* and *Q. macrolepis* from Turkey, Iran, and Israel. They were all shared with *Q. coccifera* and *Q. ilex* of the Aegean 'Cerris-Ilex' lineage. Haplotype H11 found in Iberian samples of *Q. suber* was shared with *Q. ilex* of the 'Euro-Med' lineage. Rare haplotypes restricted to a single species were H7 (one accession of *Q. cerris*), H8 (three accessions of *Q. libani*; new 'Cerris-Ilex' subtypes) and H11/H12 (eight accessions of *Q. suber*; 'Euro-Med' types); all other haplotypes were shared by more than one species of sect. *Cerris*. *Quercus cerris*, the most widespread and ecologically diverse species of sect. *Cerris*, showed the highest number of haplotypes (eight), followed by *Q. brantii* (five) and *Q. macrolepis* (four). In the latter species, haplotype H6, exclusively found in *Q. brantii*, was also found in a suspected hybrid *Q. macrolepis* × *Q. brantii* (sample ml27). All samples of *Q. ithaburensis* exhibited a single haplotype (H9). The geographically (more) restricted taxa *Q. afares*, *Q. castaneifolia*, *Q. crenata* and *Q. trojana* subsp. *euboica* (henceforth: *Q. euboica*) showed only the most frequent and widespread haplotype (H1).

In comparison (Table 2), the two East Asian members of sect. *Cerris* (*Q. acutissima* and *Q. variabilis*) displayed a higher variation at the *trnH-psbA* locus, although mostly due to indels. A higher number of haplotypes was found in these species; none of them was shared with any species of sect. *Ilex* available in gene banks (identity range: 93–99% with *Q. baroni*, *Q. dolicholepis* and/or *Q. spinosa*), and only one haplotype was shared between the two species.

No shared parsimony informative characters (PICs) were found in the East Asian samples. In contrast, three PICs were exclusively shared by haplotypes H11 and H12 (*Q. suber* from Iberian Peninsula, North Morocco). One further PIC separated H9, including all individuals of *Q. ithaburensis*, two *Q. look*, two Israeli and one Italian *Q. cerris* individuals. A single PIC also defined H4, including three co-occurring *Q. trojana* and *Q. macrolepis* accessions from the same locality in western Turkey, and another PIC was limited to two *Q. cerris* and *Q. libani* accessions from southern Turkey, corresponding to H3. Table 3 shows that the highest mean intragroup divergence in the West Eurasian dataset was found in *Q. suber* and *Q. libani*. The haplotypes of *Q. suber* and *Q. look* displayed the highest mean divergence from all other species. *Quercus variabilis* appeared more similar to its western Eurasian counterparts, while *Q. acutissima* was highly distinct.

The haplotype network (Fig. 1) shows the general coherence of the 'Cerris-Ilex' lineage, which collects haplotypes typical of the western Eurasian members of sect. *Cerris*, clearly distinct from the haplotypes found in East Asian members of sect. *Cerris* and haplotypes H11–H12 (>5 mutations separating each lineage); this latter represents a unique, early diverged plastid lineage, most frequent in the western Mediterranean populations of sect. *Ilex* (*Simeone et al., 2016*; *Vitelli et al., 2017*). Based on the relative number of mutations (1–5) separating each haplotype, the 'Cerris-Ilex' lineage can be further subdivided into two groups: (L1) a group of potentially primitive (non-derived) haplotypes (H1–H4) including the most common haplotypes (H1, H2) and still close to haplotypes found in the north-easternmost species of sect. *Ilex* (*Q. phylliraeoides*), the plastid sister lineage of 'Cerris-Ilex' (*Simeone et al., 2016*); (L2) a group of derived haplotypes (H5–H10).

Simeone et al. (2018), *PeerJ*, DOI 10.7717/peerj.5793
**Table 3  Heatmap with the mean estimates of evolutionary divergence of the trnH-psbA IGS over sequence pairs within and between the investigated taxa.**

| Dataset | Intra- | Interspecies divergence | | | | | | | | | | | |
|---|---|---|---|---|---|---|---|---|---|---|---|---|---|
| Q. afares | 0 | | 0 | 0.0001 | 0 | 0 | 0.0001 | 0.0001 | 0.0004 | 0.0009 | 0.0006 | 0.0004 | 0.0002 |
| Q. brantii | 0 | 0 | | 0.0001 | 0 | 0 | 0.0001 | 0.0002 | 0.0004 | 0.0015 | 0.0006 | 0.0004 | 0.0002 |
| Q. cerris | 0.0003 | 0.0002 | 0.0002 | | 0.0001 | 0.0001 | 0.0001 | 0.0002 | 0.0004 | 0.0008 | 0.0006 | 0.0004 | 0.0002 |
| Q, castaneifolia | 0 | 0 | 0 | 0.0002 | | 0 | 0.0001 | 0.0001 | 0.0004 | 0.0009 | 0.0006 | 0.0004 | 0.0002 |
| Q. crenata | 0 | 0 | 0 | 0.0002 | 0 | | 0.0001 | 0.0001 | 0.0004 | 0.0009 | 0.0006 | 0.0004 | 0.0002 |
| Q. trojana[b] | 0.0001 | 0.0001 | 0.0001 | 0.0003 | 0.0001 | 0.0001 | | 0.0001 | 0.0004 | 0.001 | 0.0006 | 0.0004 | 0.0002 |
| Q. ithaburensis[c] | 0.0003 | 0.0001 | 0.0003 | 0.0003 | 0.0001 | 0.0001 | 0.0002 | | 0.0005 | 0.001 | 0.0006 | 0.0004 | 0.0002 |
| Q. libani | 0.0008 | 0.0004 | 0.0004 | 0.0006 | 0.0004 | 0.0004 | 0.0005 | 0.0007 | | 0.0013 | 0.0007 | 0.0006 | 0.0005 |
| Q. look | 0 | 0.0014 | 0.0015 | 0.0014 | 0.0014 | 0.0014 | 0.0019 | 0.0018 | 0.0021 | | 0.0011 | 0.001 | 0.0009 |
| Q. suber | 0.0018 | 0.0011 | 0.0011 | 0.0013 | 0.0011 | 0.0011 | 0.0012 | 0.0011 | 0.0015 | 0.0025 | | 0.0007 | 0.0006 |
| Q. acutissima[a] | 0.0016 | 0.0008 | 0.0008 | 0.001 | 0.0008 | 0.0008 | 0.0009 | 0.0009 | 0.0013 | 0.0022 | 0.0019 | | 0.0005 |
| Q. variabilis[a] | 0.0008 | 0.0003 | 0.0003 | 0.0005 | 0.0003 | 0.0003 | 0.0004 | 0.0004 | 0.0007 | 0.0017 | 0.0014 | 0.0012 | |

**Notes.**

[a] GenBank haplotype accessions as in Table 2

[b] Including subsp. *euboica*

[c] Including subsp. *macrolepis*

Standard error estimate are shown above the diagonal.
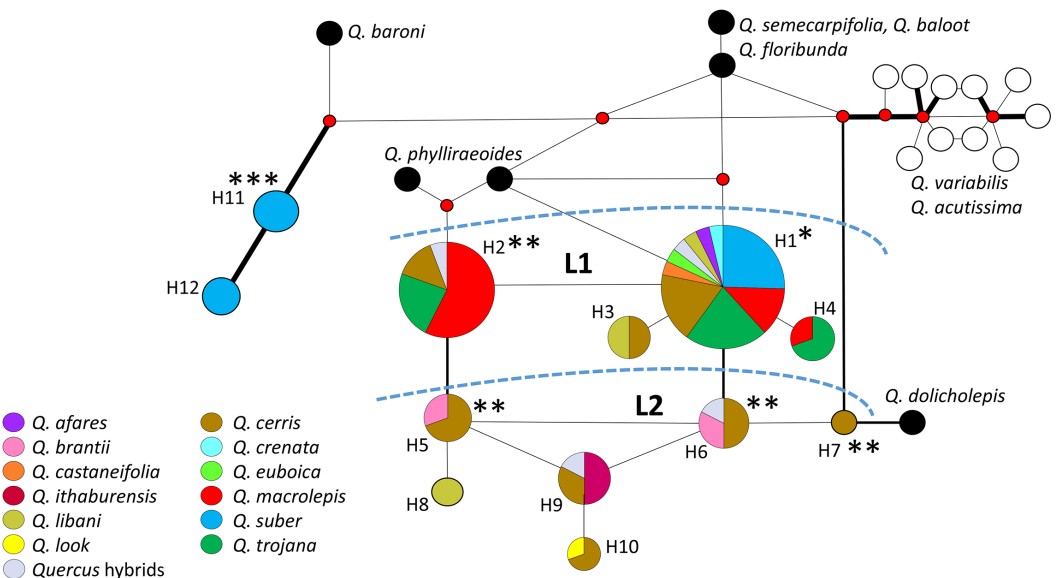

**Figure 1** **Median joining network of the *trnH-psbA* sequences in western Eurasian section *Cerris*.** Taxa are indicated with colours (see also File S1); black, Asian species of section *Ilex*; white, eastern Eurasian species of section *Cerris*. Line thickness according to 1, <5 and >5 mutations; *, shared with Asian *Ilex* oaks; **, shared with Cerris-Ilex lineage of section *Ilex*; ***, shared with West-Med lineage of section *Ilex*; L1, L2, haplotype lineages identified. All accession numbers are reported in Files S1 and S2.

Haplotypes not shared with sect. *Ilex* (H3–H4, H8–H10, H12) are derivates of the 'Cerris-Ilex' and 'Euro-Med' main types (H1–H2, H5-H7, H11), shared by both sections in the Aegean and the western Mediterranean regions. As shown in Figs. 2A–2B, plastid diversity is largely decoupled from species identity and related to geography; the least derived haplotypes within the 'Cerris-Ilex' lineage (H1 and H2) occur across the whole distribution range of the investigated group, except the Levant and the western Mediterranean (H2). All other haplotypes are more circumscribed, concentrated in Anatolia (H3–H5, H7–H8), the Levant (H6, H9–H10; the latter two showing single occurrences in Italy), and Iberian Peninsula + Morocco (H11-H12).

## Nuclear 5S rDNA diversity and species phylogeny

In contrast to *trnH-psbA*, 5S-IGS sequence variation appeared generally correlated with the taxonomy of the studied individuals, and allowed inferences on potential reticulation and inter-species relationships within the western Eurasian members of sect. *Cerris*. The 5S-IGS clones varied greatly in sequence features and length (the multiple alignment of the cloned sequences can be viewed in the Online Supplementary Archive at the journal's homepage). For instance, all *Q. brantii* clones displayed an intra-specific $(ATTT)_{1-7}$ simple sequence repeat (SSR) variation. In all the other species, this motif was either absent (replaced by a 5–12 bp long poly-T) or consisting of one to two repeats, with the exception of two clones of the suspected hybrid *Q. macrolepis* × *Q. brantii* (individual ml27) that showed four to five repetitions. Two clones of *Q. brantii* (sample br02; C. Turkey) shared a four-bp insertion with several clones of sympatric *Q. macrolepis* individuals (ml20–ml22). Three

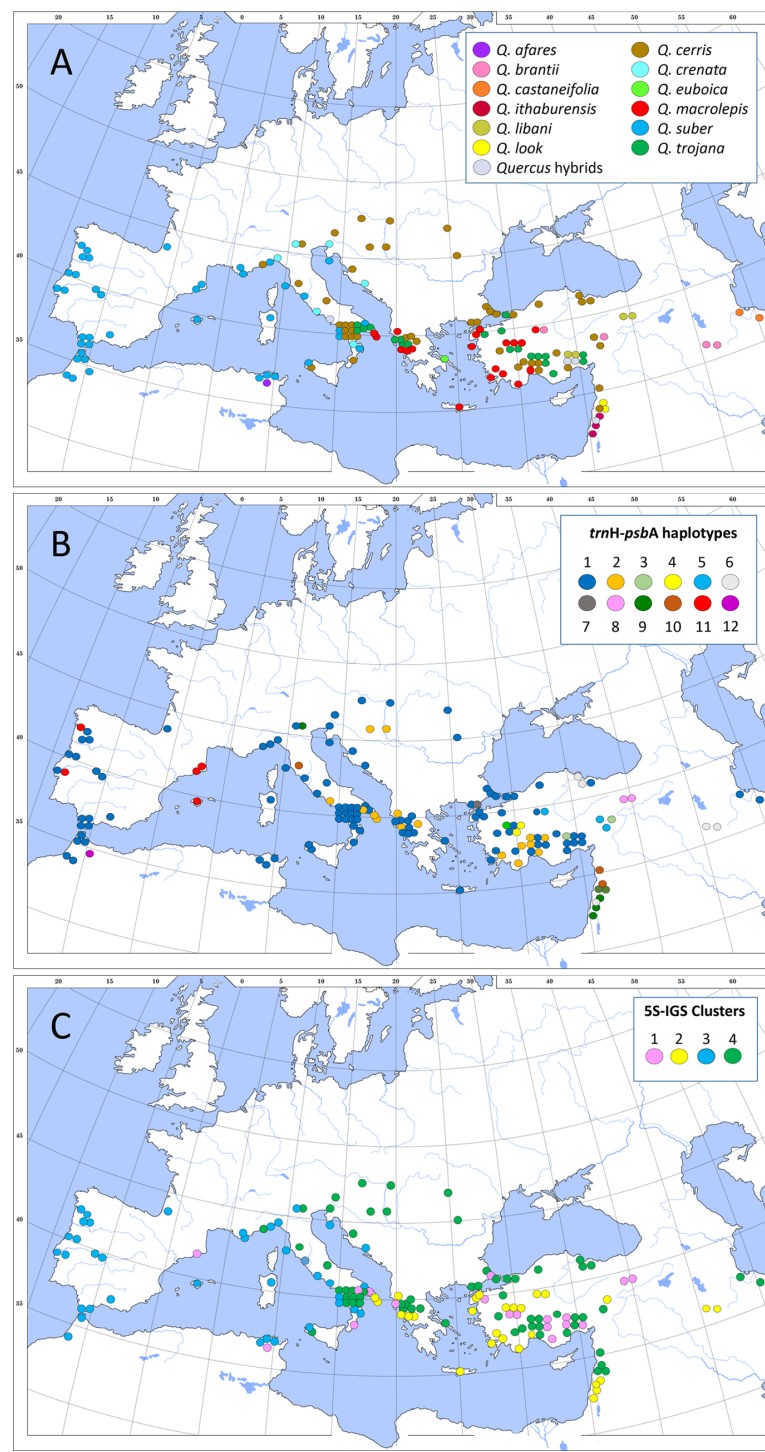

**Figure 2  Geographic representation of the investigated dataset and its molecular signatures.** (A) sample distribution. (B) *trnH-psbA* haplotypes. (C) 5S-IGS clusters; see also File S1.

*Q. libani* individuals (li02, 03, 04; S. and E. Turkey) displayed a long indel (ca. 100 bp) in (nearly) all clones ('short *libani* variant' cf. *Denk & Grimm, 2010*). The extended sample revealed that the 'short *libani* variant' is not exclusive to *Q. libani* but is rarely found also in *Q. cerris* (clones ce2104 and ce4704; Italy, W. Turkey) and *Q. trojana* (three clones of individual tj33, S. Turkey); the latter, however, is another suspected hybrid (*Q. trojana* x *Q. libani*).

Two other deletions were detected in the same region of the 'short *libani* variant'. One (22 bp) was shared by single clones of two *Q. cerris* individuals (ce18, ce22; S.W. and W. Turkey), four clones of a *Q. look* individual (lk2; Israel) and two clones of *Q. trojana* (individual tj40; S. Turkey). The second (~100 bp), largely overlapping with the deletion of the '*short libani variant*', but beginning a few basepairs downstream, was shared by one clone of *Q. cerris* (ce34; N. Turkey) and one clone of *Q. macrolepis* (ml26; S. Greece). An 8-bp deletion occurred exclusively in *Q. suber* and *Q. crenata*, with the exception of single clones of samples su07, su09 (N.E. and S. Spain), su37 (Croatia), su53 (S. Italy), cr02 (C. Italy), two clones of sample cr04 (Slovenia) and cr06 (N.E. Italy), and three clones of sample cr05 (Croatia). The same deletion also occurred in two clones of sample tj08, a *Q. suber* × *Q. trojana* cultivation hybrid. Further deletions (1–60 bp) were scattered along the alignment and found only in single individuals (e.g., it04, Israel; ml10, N.W. Greece). Finally, an 18-bp highly variable region was exclusively found in some clones of four co-occurring *Q. trojana* samples (tj03–05, tj16; S.C. Turkey).

The main diversity values of the investigated dataset are reported in Table 4. Identical 5S-IGS sequences typically occur in the same individual and species, and, to a lesser extent, in sympatric, different species (e.g., *Q. brantii*, *Q. cerris*, *Q. trojana*, *Q. look*; see also File S3). On the contrary, *Q. afares*, *Q. castaneifolia*, *Q. libani*, *Q. ithaburensis*, *Q. macrolepis* and *Q. euboica* showed high species coherence. *Quercus suber* and *Q. macrolepis* showed the highest number of intra-individual and intra-specifically shared clones, whereas *Q. cerris, Q. trojana* and *Q. ithaburensis* displayed the highest levels of unique variants. No variants were shared between *Q. trojana* and *Q. euboica*; *Q. suber* and *Q. crenata* (but not *Q. cerris*) shared 69 identical sequences and are the genetically most similar taxon pair. The pairwise uncorrected *p*-distance range of the total dataset was much higher than for the plastid marker (0–0.209), with highest values scored by *Q. cerris* and *Q. trojana*. The mean intra-specific molecular diversity estimated within sequence pairs (Table 5) was lowest in the two narrow endemics *Q. afares* and *Q. castaneifolia* and highest in *Q. brantii* and *Q. ithaburensis*. Across the entire dataset, *Q. brantii, Q. macrolepis* and *Q. ithaburensis* were the most diverging taxa; the least divergent being *Q. afares* and *Q. castaneifolia*. The mean divergence value between *Q. macrolepis* and *Q. ithaburensis* (0,0376), treated as subspecies of *Q. ithaburensis* in current regional floras, was similar to values detected between these taxa and the other species (e.g., *Q. afares, Q. brantii, Q. libani*). Likewise, the divergence recorded between the putative conspecific *Q. trojana* and *Q. euboica* (0.0266) was comparable to the estimates calculated between these and other taxa (e.g., *Q. afares*, *Q. cerris, Q. look, Q. libani*). The putative hybrid taxon *Q. crenata* displayed the lowest divergence (0.0197) with *Q. suber,* one of the assumed parental species, and a slightly higher

**Table 4 Diversity values of the 5S IGS clones in the investigated dataset.**

| Dataset | N | Cs | L | O (u/i/a/s) | D | *p* | C |
|---|---|---|---|---|---|---|---|
| West Eurasian Cerris oaks | 194 | 856 | 427 | 457/186/121/79 | | 0.000–0.209 ± 0.021 | 1–4 |
| *Q. afares* | 5 | 17 | 379 | 10/2/5/0 | | 0.000–0.019 ± 0.006 | 1 |
| *Q. brantii* | 7 | 26 | 403 | 9/11/4/2 | *Q. cerris* (3)/*Q. look, Q. suber* (1) | 0.000–0.088 ± 0.014 | 4 |
| *Q. castaneifolia* | 2 | 2 | 375 | 2/0/0/0 | | 0.005 ± 0.003 | 2 |
| *Q. cerris* | 48 | 207 | 392 | 157/21/24/5 | *Q. brantii* (1), *Q. trojana* (1), *Q. suber* (1) | 0.000–0.202 ± 0.02 | 1[c], 2 |
| *Q. crenata* | 6 | 29 | 387 | 19/4/2/4 | *Q. suber* (65) | 0.000–0.054 ± 0.012 | 2[d], 3[e] |
| *Q. libani* | 5 | 20 | 382 | 13/3/4/0 | | 0.000–0.040 ± 0.009 | 1, 2[f] |
| *Q. look* | 3 | 14 | 383 | 10/3/0/1 | *Q. brantii/Q. cerris* (3) | 0.000–0.032 ± 0.009 | 2 |
| *Q. macrolepis*[a] | 28 | 158 | 402 | 44/71/43/0 | | 0.000–0.065 ± 0.013 | 4 |
| *Q. ithaburensis* | 5 | 21 | 388 | 15/6/0/0 | | 0.000–0.079 ± 0.014 | 4[g] |
| *Q. suber* | 38 | 153 | 385 | 30/48/8/67 | *Q. brantii* (1), *Q. cerris* (1), *Q. crenata* (4) | 0.000–0.168 ± 0.019 | 1[h], 3 |
| *Q. trojana*[b] | 43 | 192 | 391 | 130/30/31/1 | *Q. cerris* (1) | 0.000–0.198 ± 0.020 | 1, 2, 3[i] |
| *Q. euboica* | 4 | 17 | 382 | 17/0/0/0 | | 0.000–0.059 ± 0.012 | 2 |

**Notes.**

N, number of individuals; Cs, number of clone sequences; L, Aligned length (bp); O, occurrence of the IGS variants (u, unique, i, intra-individually identical; a, intra-specifically shared; s, inter-specifically shared); D, distribution of the interspecifically shared variants (no. of variants); *p*, uncorrected *p*-distance range (STD); C, clusters identified with the neighbour-net analyses.

[a]Including one putative hybrid with *Q. brantii*.
[b]Including putative hybrids with *Q. suber* and *Q. libani*.
[c]Sample ce50 (S Italy).
[d]Sample cr04 (Slovenia).
[e]Including odd-placed sample cr05 (Croatia).
[f]Sample li01 (S Turkey).
[g]Including odd-placed sample it03 (Israel).
[h]Sample su09 (S Spain).
[i]Sample tj08 (Botanical Garden of Naples).

estimate (but similar to the values scored with other taxa, e.g., *Q. afares*, *Q. castaneifolia*, *Q. look*) with *Q. cerris* (0.0266), the other putative parental species.

The clone-based ML tree rooted on *Q. baloot* and *Q. floribunda* (West-Asian members of sect. *Ilex*) showed four main topological features (grades/clades) generally coherent with taxonomy (Fig. 3, see also File S4). These grades/clades collected to a large degree clones of (1) *Q. crenata* and *Q. suber* (resolved as proximal, weakly differentiated grade), (2) *Q. brantii*, *Q. ithaburensis* and *Q. macrolepis* (the most highly supported clade: BS$_{ML}$ = 84), (3) *Q. trojana* (a large heterogeneous grade), and (4) *Q. cerris* (the distal, terminal, clade with diminishing support). *Quercus libani* clones (short and normal-length variants) were present in all clades/grades except grade 1. A moderately supported clade (BS = 63) including all *Q. afares* clones was placed as sister to the main clade including clades/grades 2–4; *Q. castaneifolia* clones were placed within grade 3. Clones of *Q. ithaburensis* also occurred in grade 3, *Q. brantii* and *Q. crenata* in clade 4, *Q. look* and *Q. euboica* in grade 3 and 4. A few clones of *Q. cerris* and *Q. trojana* occurred scattered across the tree (often in proximal positions).

Of the three clones sequenced from individual ml27, a suspected *Q. macrolepis* × *brantii* hybrid, one was identical to another *Q. macrolepis* clone (individual ml08) and the other two clustered together with *Q. brantii*. Likewise, three and two of the five clones sequenced

Simeone et al. (2018), *PeerJ*, DOI 10.7717/peerj.5793

Peer**J**

**Table 5  Heatmap with the mean estimates of evolutionary divergence of the nuclear 5S IGS over sequence pairs within and between the investigated taxa.** Standard error estimates are shown above the diagonal.

| Dataset | Intra- | *Q. afares* | *Q. brantii* | *Q. cerris* | *Q. castaneifolia* | *Q. crenata* | *Q. euboica* | *Q. ithaburensis* | *Q. libani* | *Q. look* | *Q. macrolepis* | *Q. suber* | *Q. trojana* |
|---|---|---|---|---|---|---|---|---|---|---|---|---|---|
| | | | | | | | Intergroup divergence | | | | | | |
| *Q. afares* | 0.0057 | | 0.0065 | 0.0059 | 0.0058 | 0.0048 | 0.006 | 0.0064 | 0.0058 | 0.0062 | 0.0085 | 0.0052 | 0.0049 |
| *Q. brantii* | 0.0357 | 0.0353 | | 0.0068 | 0.0066 | 0.006 | 0.0067 | 0.0044 | 0.0063 | 0.0068 | 0.0054 | 0.006 | 0.0061 |
| *Q. cerris* | 0.0167 | 0.0244 | 0.0427 | | 0.0042 | 0.0043 | 0.0021 | 0.0062 | 0.0046 | 0.0021 | 0.0088 | 0.0056 | 0.0032 |
| *Q. castaneifolia* | 0.0053 | 0.0169 | 0.0343 | 0.0166 | | 0.0048 | 0.004 | 0.0062 | 0.0045 | 0.0041 | 0.0085 | 0.0055 | 0.0035 |
| *Q. crenata* | 0.023 | 0.0241 | 0.0422 | 0.0266 | 0.0216 | | 0.0044 | 0.0056 | 0.005 | 0.0046 | 0.0078 | 0.0028 | 0.004 |
| *Q. euboica* | 0.0194 | 0.026 | 0.0436 | 0.0193 | 0.0171 | 0.0285 | | 0.0061 | 0.0046 | 0.0026 | 0.0088 | 0.0055 | 0.0031 |
| *Q. ithaburensis* | 0.0367 | 0.0364 | 0.0386 | 0.0423 | 0.0344 | 0.0422 | 0.0431 | | 0.0057 | 0.0062 | 0.0056 | 0.0058 | 0.0056 |
| *Q. libani* | 0.0166 | 0.0232 | 0.0382 | 0.0244 | 0.0179 | 0.028 | 0.0255 | 0.037 | | 0.0046 | 0.0078 | 0.0054 | 0.0041 |
| *Q. look* | 0.0134 | 0.023 | 0.0406 | 0.0159 | 0.0139 | 0.0255 | 0.0175 | 0.0401 | 0.0214 | | 0.0089 | 0.0057 | 0.0032 |
| *Q. macrolepis* | 0.0194 | 0.0399 | 0.036 | 0.0498 | 0.0399 | 0.0473 | 0.0507 | 0.0376 | 0.0434 | 0.0481 | | 0.0079 | 0.0079 |
| *Q. suber* | 0.0135 | 0.0211 | 0.0382 | 0.0279 | 0.0201 | 0.0197 | 0.0288 | 0.0385 | 0.0262 | 0.026 | 0.0423 | | 0.0045 |
| *Q. trojana* | 0.027 | 0.0257 | 0.0433 | 0.0259 | 0.0195 | 0.0304 | 0.0266 | 0.043 | 0.0259 | 0.0235 | 0.0492 | 0.0287 | |

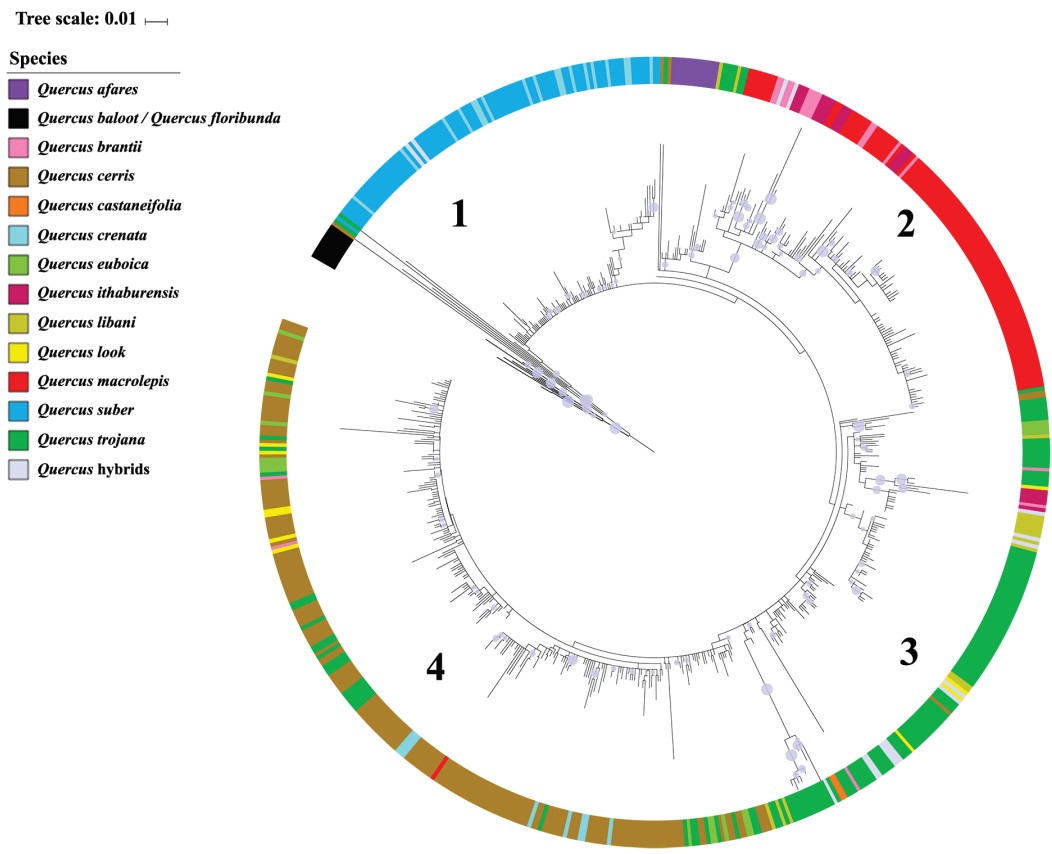

**Tree scale: 0.01**

**Species**

- ■ *Quercus afares*
- ■ *Quercus baloot / Quercus floribunda*
- ■ *Quercus brantii*
- ■ *Quercus cerris*
- ■ *Quercus castaneifolia*
- ■ *Quercus crenata*
- ■ *Quercus euboica*
- ■ *Quercus ithaburensis*
- ■ *Quercus libani*
- ■ *Quercus look*
- ■ *Quercus macrolepis*
- ■ *Quercus suber*
- ■ *Quercus trojana*
- ■ *Quercus hybrids*

**Figure 3  5S-IGS Clone-based RaxML tree.** The tree was tentatively rooted on *Q. baloot* and *Q. floribunda*, two western Asian oaks of Sect. *Ilex* (cf. *Denk & Grimm, 2010*; *Simeone et al., 2016*). Colours as in Figs. 1, 2A and File S1. Numbers 1–4 indicate the four major clades identified. Branch bootstrap support (1–100) is scaled as circles of increasing size (see also File S4 for details on clone labels and bootstrap values).

in sample tj08, a *Q. trojana × Q. suber* hybrid, clustered within the respective parental subtrees; the same applies to the five clones of the sample tj33, a supposed *Q. libani × Q. trojana* hybrid. Conversely, all the three clones sequenced in sample tj02, another tree determined as possible *Q. libani × Q. trojana* hybrid, clustered with *Q. trojana*.

The networks based on transformed 5S-IGS data (Fig. 4 based on AVG-transformed uncorrected distances; Fig. 5 based on PBC-transformed distance matrix; only individuals represented by more than four clones included) largely confirmed the earlier found intra- and inter-species relationships (*Denk & Grimm, 2010*; because of the amount of shared identical clones, the MIN-transformed networks are largely collapsed, but included in the Online Supporting Archive). Four clusters can be observed: Cluster 1, the 'oriental' lineage of sect. *Cerris*, is the least coherent cluster and equivalent to a grade in a corresponding outgroup-rooted (*Q. baloot, Q. floribunda*) tree. This lineage included, in the AVG network, *Q. afares*, four out of five *Q. libani* individuals and about half of the *Q. trojana* individuals (Fig. 4). Its counterpart, Cluster 2, the 'occidental' lineage, accomodated all *Q. look,*

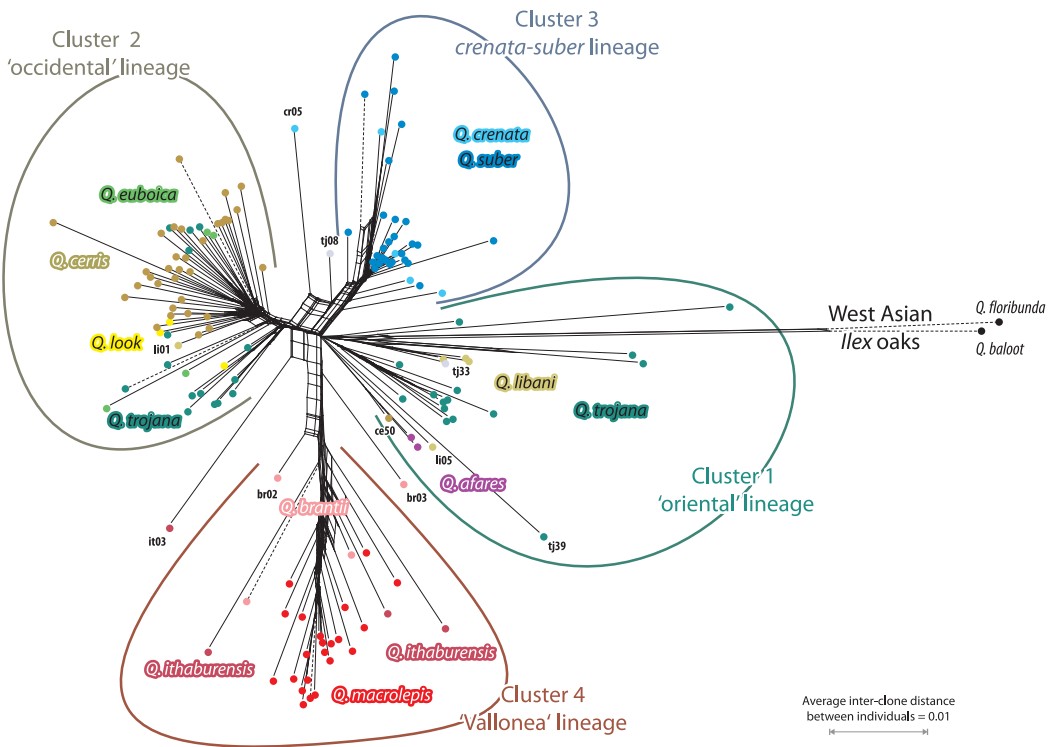

**Figure 4** **Network based on transformed 5S-IGS data showing inter-individual average (AVG) clonal distance relationships.** Only individuals represented by more than four clones are included (reconstructions for other cut-offs, *m* = 2, 3, or 5, are included in the File S5; see also our Online Supplementary Archive).

*Q. euboica,* the remaining *Q. trojana* and *Q. libani* samples, and all but one *Q. cerris* individual (ce50; Figs. 4 and 5). The PBC network (Fig. 5) reveals a more gradual shift between these two clusters, with *Q. afares* splitting off with two genetically similar *Q. cerris* and *Q. libani* individuals (ce50, li05). The reason for this is that the PBC transformation has a higher chance to capture evolutionary signals (*Göker & Grimm, 2008*). Cluster 3 included *Q. suber* and *Q. crenata*. Here, the only difference is the boxyness inflicted by individuals cr05 and tj08. Cluster 4 included the 'Vallonea' (or *Aegilops*) oaks, *Q. brantii*, *Q. ithaburensis* and *Q. macrolepis*, with two *Q. brantii* individuals (br02 and br03, with diverging 5S-IGS features and variants; File S4) in proximal (br02 in Figs. 4 and 5; br03 in Fig. 4) or off-cluster (br03 in Fig. 5) position. Thus, the basic structure of the AVG and PBC networks and the ML tree are equivalent, but they differ in placing the outgroup taxa, and the networks refine inter-species relationships.

The AVG network (Fig. 4) better captured putative hybrid and intrograded individuals. Strong ambiguous signals came from the hybrid *Q. trojana* × *Q. suber* (tj08), one *Q. crenata* individual (cr05, terminals in the box-like structure connecting the 'occidental', cluster 2, and *crenata-suber* lineage, cluster 3), and one *Q. ithaburensis* individual (it03, terminal in the box-like structure between clusters 2 and 4). The placement of one *Q. libani* individual (li01, inserted in cluster 2), with normal-long variants in the clone sample, and one *Q. cerris*

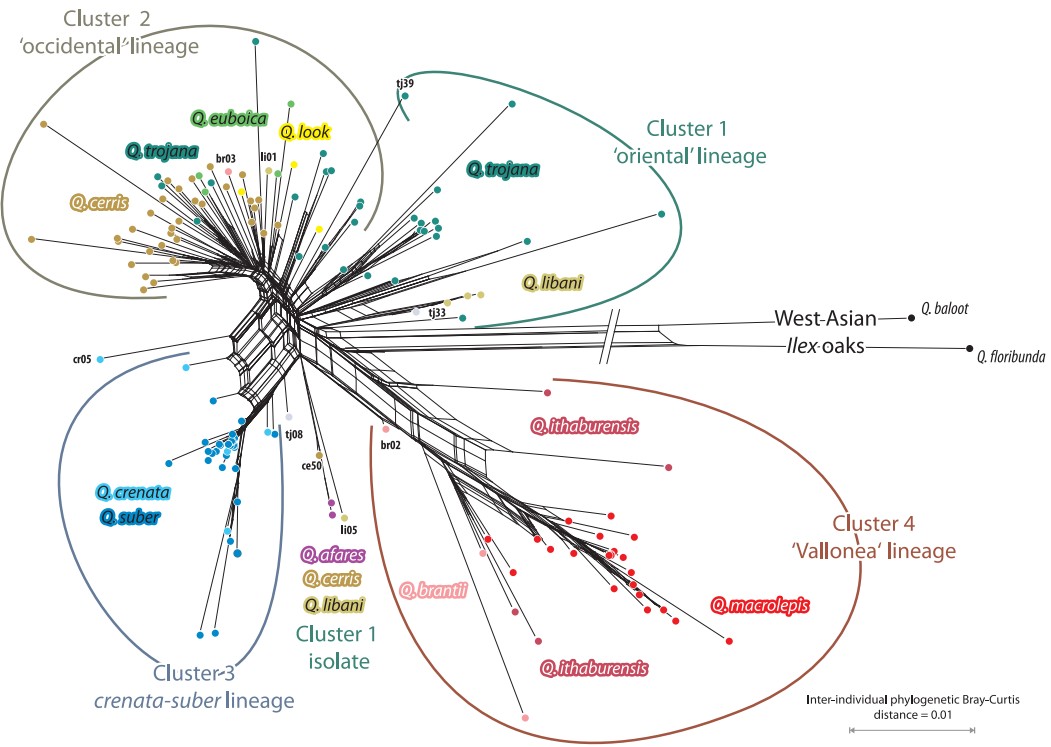

**Figure 5** **Network based on transformed 5S-IGS data showing inter-individual PBC clonal distance relationships.** Only individuals represented by more than four clones are included (reconstructions for other cut-offs, m = 2, 3, or 5, are included in our Online Supplementary Archive).

(ce50, in cluster 1 close to the *Q. afares* subgroup; cf. Fig. 5) does not follow the general trend. Long terminal edges indicative of unique individual clone samples (combinations) are found in each cluster. Besides the outgroup *Q. baloot* and *Q. floribunda*, these samples include individuals of *Q. cerris* (ce29, 44), *Q. trojana* (tj03, 16, 24, 39, 45), *Q. suber* (su07, 29, 49), *Q. brantii* (br06), *Q. ithaburensis* (it04, 05), and *Q. macrolepis* (ml10). Some of these samples had unique deletions or highly divergent regions in their clones (e.g., it04, ml10, tj03, tj16; see above). The networks produced with individuals represented by ≥2, ≥3 (and ≥5) clones did not change this structure (File S5); they allowed inclusion of all individuals into the four clusters matching the general scheme and pinpointed a few other (possible) exceptions. *Quercus castaneifolia* (represented by two clones) and one sample of *Q. crenata* (cr04; three clones) formed part of the 'occidental' lineage, cluster 2; one sample of *Q. suber* (su09; three clones) was included in cluster 1, the 'oriental' lineage of Sect. *Cerris* (see Table 4); sample br01 (three clones) was placed at the root of cluster 4, similarly to samples br02 and br03. The geographical distribution of the four clusters is shown in Fig. 2C.

In contrast, the PBC network (Fig. 5) provided a better basis for inferring the evolution and differentiation (speciation) processes. The 'oriental' (cluster 1) and 'occidental' lineages are clearly connected and form a continuum, with the easternbound *Q. trojana* and *Q. libani* representing a diverged, differentiated pool from which the other species and western *Q.*

*trojana* derived. The western Mediterranean *crenata-suber* lineage is clearly different and only linked to the main pool by occasional introgression or hybridisation with nearby members of the 'occidental' lineage (in nature: *Q. cerris*). The same holds even more for the 'Vallonea' oaks (Cluster 4), which appear to have split before the remainder of western Eurasian *Cerris* (but long-branch/-edge attraction with the extreme long-edged outgroup needs to be considered). A clear signal in the PBC network (Fig. 4) is the uniqueness of *Q. afares*, a disjunct outpost of the putative 'oriental' lineage, genetically closely related to geographically very disjunct (C./S. Anatolian) individuals of *Q. cerris* and *Q. libani* (cf. Fig. 3 showing a *Q. afares* subclade, and File S4, same tree with clones labelled).

## DISCUSSION

The western Eurasian members of sect. *Cerris* exhibit a *trnH-psbA* diversity well comparable with the Mediterranean oaks of sect. *Ilex* (*Vitelli et al., 2017*) and Fagaceae in general (*Simeone et al., 2016*). As discussed in *Grímsson et al. (2016)*, the plastid genealogy in this genus is largely decoupled from species identity. Nevertheless, the strong geographic signal of plastid data provides useful information to decipher population-area relationships and taxon histories (e.g., isolation, reticulation, introgression; cf. *Pham et al., 2017*). Conversely, the intergenic spacers of the 5S rDNA were confirmed as the most variable nuclear gene region for a large range of plants (*Volkov et al., 2001*; *Forest et al., 2005*; *Lehtonen & Myllys, 2008*; *Denk & Grimm, 2010*; *Grimm & Denk, 2010*). They were highly variable across the entire dataset and displayed inter-individual patterns that allowed circumscription of most of the investigated species; the intra-individual variation in the 5S-IGS further helped to recognize hybridization and infer other reticulation events such as introgression. Insights gained from the combination of data sources were concordant with the known ecology and biogeography of the studied taxa. So far, all other gene regions sequenced in oaks showed (much) less divergence at the intra- and interspecies level (ITS: *Denk & Grimm, 2010*; *LEAFY* intron: *Oh & Manos, 2008*; various single-copy nuclear genes: *Hubert et al., 2014*; various plastid intergenic spacers and genes: *Simeone et al., 2013*). Increased resolution may be achieved by using phylogenomic data (e.g., *McVay, Hipp & Manos, 2017*) but it would be more laborious and cost-intensive for large intra- and interspecific data sets.

### Molecular recognition of species and species diversity in *Quercus* section *Cerris*

Widespread species such as *Q. cerris* and (to a lesser extent) *Q. libani* (Table 1) showed the highest plastid diversity in terms of number of haplotypes and parameters of molecular differentiation (Tables 2 and 3); *Q. suber* diversity was inflated by the occurrence of few divergent haplotypes linked to—and possibly captured from—the 'Euro-Med' lineage of sect. *Ilex*.

The strikingly high haplotype richness of *Q. cerris*, especially in the eastern part of this species' range (cf. *Bagnoli et al., 2016*), mirrors the high morphological plasticity of this oak (many different varieties and subspecies are traditionally reported; for example, IOPI lists 30 *formae* and 17 varieties) and its ecological adaptability. Likewise, *Q. cerris* also displayed a high nuclear (5S-IGS) diversity (Table 4, Fig. 3). This oak has the largest range and broadest

climatic envelope (from perhumid *Cfa, Cfb* via summer-dry warm temperate climates to *BSk*) and it is the only species of sect. *Cerris* naturalized on the British Islands (*Cfb*) and cultivated all over continental Europe (mostly *Cfb*, sheltered *Dfb*). Indeed, establishment of a large range across the geologically and ecologically dynamic West Eurasian region might have provided many opportunities for diversification, isolation, drift, conservation of variants and eventual reticulation with sibling species.

*Quercus suber*, instead, displayed the lowest number of unique 5S-IGS variants (like *Q. macrolepis*) with a low diversity (Tables 4 and 5), which might indicate ongoing genetic erosion (possibly due to the species domestication for cork, tannins, wood and fruits exploitation). In view of the high species-coherence detected in the other conspecific samples, the few Iberian samples not following the general trend may indicate introgression of *Q. suber* into *Q. ilex* (individuals with 'Euro-Med' plastid haplotype H11) or reflect ancient reticulation and retention of ancestral signatures (individuals with the widespread, putatively ancestral H1 haplotype).

In *Q. libani*, the high haplotype diversity coincides with a moderate diversity at the 5S-IGS locus, characterized by interesting variation among the cloned sequences. Two individuals (li01, li05) show potential introgression (e.g., with sympatric *Q. trojana* samples tj05, tj35; Files S1 and S4). Alternatively, they might represent an ancestral line of diversification within the section (the short 5S-IGS variant so far only known for *Q. libani*, see *Denk & Grimm (2010)*, also occurs in clones of two non-sympatric *Q. cerris* individuals). Together with *Q. brantii, Q. libani* is the easternmost oak among the western Eurasian *Cerris*, and it is extremely variable at the morphological and ecological level (occurring in climates ranging from *Csa* to *Dsb*). For instance, *Djavanchir-Khoie (1967)* described up to 12 intra-specific taxa within *Q. libani*, and introgression phenomena with other co-occurring oaks have been postulated (*Menitsky, 2005*; *Khadivi-Khub et al., 2015*). Likewise, in the nearby region of Iranian Kurdistan, this oak showed three distinct gene pools based on nuclear microsatellites (*Khadivi-Khub et al., 2015*).

Based on the phylogenetic reconstructions, the 5S-IGS sequence diversity detected in *Q. trojana* outmatched the high levels displayed by *Q. cerris*, rendering it the genetically least-coherent species of the section. Samples with highly diverse clones were detected in both species (see 'Results'), but individuals of *Q. trojana* can be found in two different clusters in the 5S-IGS network. Reflecting its more limited distribution and climatic niche (*Csa, Csb*), the haplotype diversity is substantially lower in *Q. trojana* s.l. (*Q. trojana* + *Q. euboica*) than in *Q. cerris* (Tables 2 and 3). This finding indicates that the two species retain exceptionally high intra- and inter-individual variability, possibly conserving ancestral variants lost in more homogenized and/or geographically restricted species of sect. *Cerris*, and corresponds, at the plastid level, to the relative extension of their geographic (and ecological) ranges. Accordingly, the two endemic taxa *Q. afares* and *Q. castaneifolia* appeared the least diverse (Tables 4 and 5), although more data are needed for *Q. castaneifolia*, which was here represented by only two samples. The same holds true for the two other taxa with narrow ranges in our dataset, *Q. euboica* and *Q. look*. Both are characterized by low levels of genetic diversity (Tables 4 and 5); however, despite the

low number of individuals investigated, our results allow first taxonomic inferences in both cases.

*Quercus euboica* appears genetically isolated from *Q. trojana*, based on the number of unique 5S variants (Table 4) and the relative inter-taxa divergence (Table 5). This oak grows isolated from *Q. trojana* on the Greek island of Euboea and differs morphologically by its coriaceous leaf texture and the conspicuous white tomentum of the abaxial leaf surface that is made up of stellate trichomes (T Denk, pers. obs., 2008 and 2017). In addition, *Q. euboica* is characterized by special edaphic conditions, growing on serpentine rocks. All these data indicate that the Euboean oak should be better considered as an independent species requiring special protection. Another (hairy) variant of *Q. trojana* has been locally described at the south-eastern margin of the species' range, in South-central Turkey (*Q. trojana* subsp. *yaltirikii; Zielinski, Petrova & Tomaszewski, 2006*). Some samples collected in the nearby area (tj03, 04, 05, 16) showed 5S-IGS clones with a unique, highly divergent motif, and grouped (mostly) in a specific sublcade (Fig. 3, File S4). However, more (morpho-ecological) data are needed to implement the description of *Q. trojana* in this part of its range.

The *Q. look* samples showed a distinct plastid-nuclear signature combination (Tables 2–4; File S4) linking it with *Q. cerris* (Figs. 3–5). In addition, this species showed the lowest mean estimate of evolutionary divergence of the nuclear 5S IGS together with *Q. castaneifolia* (Table 5). Although the taxonomic rank of this rare, enigmatic taxon cannot be yet established with certainty (a hybrid origin or a local diversification of an ancestral form of *Q. cerris* seem equally probable), the two previous assessments of this oak as synonym of *Q. ithaburensis* or *Q. ithaburensis* × *Q. libani* hybrid (Table 1) can be rejected. Additional investigations are required to evaluate if the morphology of *Q. look* justifies its exclusion from the genetically and morphologically variable *Q. cerris*.

Finally, a distinct group with medium-high levels of plastid and nuclear diversity includes *Q. brantii, Q. ithaburensis,* and *Q. macrolepis* (Tables 2, 4 and 5; 'Aegilops oaks') with a range centred in the *Csa* climates of the central-eastern Mediterranean region, providing a low-land analogue to the situation in *Q. trojana-euboica-libani*. The Aegilops oaks are a highly specialized group, morphologically and ecologically well distinct from the other oaks in the *Cerris* section (*Menitsky, 2005*). The detected genetic diversity at the plastid and, especially, the nuclear markers (Table 5) clearly indicates the genetic isolation from the rest of the *Cerris* oaks and progressive inter-specific differentiation. Geographic, morphological and ecological differences are also evident in *Q. ithaburensis* and *Q. macrolepis* (*Dufour-Dror & Ertas, 2002; Dufour-Dror & Ertas, 2004*). On these grounds and considering the high inter-taxon 5S-IGS divergence supported by the different haplotypes (Tables 2–4), we suggest these two forms be treated as separate species (cf. *Denk et al., 2017*, appendix: http://dx.doi.org/10.1101/168146). Interestingly, *Q. brantii* appeared as the most diverse of the three taxa (Tables 4 and 5), and displayed some 5S-IGS variants shared with *Q. cerris* and *Q. suber* (File S3), which might indicate the occurrence of ancestral traits.

## Hybrid detection within *Quercus* Section *Cerris*

Besides the deletion in *Q. libani*, some other sequence features, typical of other species (e.g., the SSR motif in *Q. brantii*, the deletion in *Q. suber*), confirmed the hybrid identity of a few samples included in our dataset (sample ml27, tj08 and tj33, supposed hybrids *Q. macrolepis* × *brantii*, *Q. trojana* × *Q. suber* and *Q. trojana* × *Q. libani*, respectively). The haplotypes of these samples (ml27: H6, exclusive of *Q. brantii*; tj08: H2, never found in *Q. suber*), also allowed identification of the maternal species. This finding confirms that these oaks can occasionally hybridize in sympatry, and evidence for such hybridization is found in the nuclear genome (see also *Fitzek et al., 2018*).

Further instances of hybridization and/or introgression events could be inferred from common sequence features, inter-specifically shared variants (see 'Results' section, Table 4, Fig. 3, File S3), and the placement of individuals in the AVG Neighbour-Net (Fig. 4), mostly involving Anatolian samples of *Q. cerris*, *Q. brantii*, *Q. libani*, *Q. macrolepis* and *Q. trojana*. However, in many cases the involved individuals grew hundreds to a few thousands of kilometres from each other. Based on their (relative) spatial proximity, introgressions could be suggested for samples *Q. brantii* (br02; C. Anatolia) and *Q. macrolepis* (e.g., ml20–22; W. and S. Anatolia) sharing sequence features, and South Anatolian *Q. libani* (li01) and *Q. trojana* (e.g., tj05, tj35) sharing sequence features and *trnH-psbA* haplotypes. Outside Anatolia, evidence of reticulation (shared variants) can be traced between Israelian *Q. cerris* and *Q. look* (sample ce38, lk03) and between Balkan *Q. cerris* and *Q. suber* (sample ce43, Serbia; su37, Croatia). Introgression and past hybridization events between all these species or their precursors is a possible explanation. At the same time, retention of ancestral traits cannot be discarded, as *Q. cerris* (5S-IGS, *trnH-psbA*) and *Q. trojana* (5S-IGS) cover most variability found in sect. *Cerris* and are highly variable, especially in Anatolia.

In this context, the unresolved taxonomic status of *Q. crenata* can be discussed in view of the present results (Figs. 3–5). The species is more closely related to *Q. suber* than to *Q. cerris* and part of the distinct *Q. crenata-suber* lineage. The relative distribution of the clones (in the *Q. cerris-* and in the *Q. suber*-dominated clades) and the large extent of variants shared with *Q. suber* can be interpreted as evidence of co-existing both (1) *Q. cerris* × *Q. suber* F1 hybrids and (2) introgressive forms into either *Q. cerris* (North East Italy/Balkans) or *Q. suber* (Italian peninsula), in partial agreement with *Conte, Cotti & Cristofolini (2007)*. However, all forms would look phenotypically quite similar (intermediacy of habitus, leaf and bark shape between *Q. suber* and *Q. cerris* is traditionally used as a diagnostic character of *Q. crenata*), which seems inconsistent with their presumable different genome composition. Clearly, the hybrid/introgressed phenotypes can be affected by several phenomena such as segregation, epistasis, heterosis, and maternal origin (*Rieseberg & Ellstrand, 1993*). At the same time, we note that the diagnostic traits used for *Q. crenata* occur in other species of sect. *Cerris* (corky bark: *Q. afares*, *Q. variabilis*, various forms of *Q. cerris*; *Menitsky, 2005*; semi-evergreen habitus: *Q. trojana* and *Q. libani*; *Yaltırık, 1984*; crenate leaves: part of the morphological variation of *Q. cerris* and *Q. suber*). Also, *Q. cerris* shares plastid and nuclear signatures with other species of sect. *Cerris,* including geographically isolated, morphologically distinct taxa such as *Q. afares*, *Q. castaneifolia*, *Q. euboica* (traditionally included in *Q. trojana*)*,* and *Q. look* (traditionally included in

*Q. ithaburensis*), which are part of the 'occidental' lineage (see 'Results'). Besides occasional hybridizations (sample cr05; Fig. 4), the alternative explanation is that *Q. crenata* represents a less-derived species possessing a limited gene pool within an autonomous evolutionary lineage including *Q. suber* (Fig. 5). Being closer to the common root, it retained imprints of common origin, possibly ancient reticulation, with the (proto-)*Q. cerris* ('occidental') lineage, representing a geographic-evolutionary gradient ('oriental' lineage → 'occidental' lineage → *crenata-suber* lineage). *Quercus crenata* may then just represent the remainder of the ancestral form from which *Q. suber* evolved rather than being the product of secondary contact between *Q. cerris* and *Q. suber*.

We also found no evidence to support the hybrid origin of *Q. afares* (*Q. suber* x *Q. canariensis*; the latter is a member of sect. *Quercus*) as suggested by *Mir et al. (2006)* based on cpDNA-RFLP and allozymes (cf. *Welter et al., 2012*; *Mhamdi et al., 2013*). 5S-IGS variants and plastid signatures of western Eurasian white oaks ('roburoid oaks' in *Denk & Grimm, 2010*; see also *Simeone et al. 2016*, Fig. 1) are very distinct from *Cerris* types and should be detectable unless the F1 hybrids, with *Q. suber* as a maternal parent, only backcrossed with the local *Q. suber* but not *Q. canariensis.* However, genetic exchange with local *Q. suber* can be excluded, since no *Q. suber*-typical 5S-IGS variants, or obvious phylogenetic linkage with the *Q. crenata-suber* lineage, were found in *Q. afares.* Ongoing next-generation target sequencing of the 5S-IGS region (producing several 10,000 5S-IGS sequences per sample/individual) showed, so far, no evidence for a clone-sampling artefact in the studied individuals of *Q. afares.* The possibility of incomprehensive clone-sampling can thus be discarded. Analoguous to *Q. crenata,* the ancestry level of *Q. afares* may explain earlier findings interpreted towards a hybrid origin: being much closer to the common ancestor of sect. *Cerris* than *Q. suber*, this species may have retained (some) genetic imprints today found in members outside its section. This would explain also the association of *Q. afares* with two other, geographically very distant, individuals of the 'oriental' lineage (ce50, li05, the only *Q. libani* without the 'short' *libani* 5S-IGS variants but showing variants similar to Anatolian *Q. trojana*).

Aside from putative hybrid species and swarms (see also the ambiguous placement of sample it03 the 5S-IGS network), our data demonstrates a general permeability of species boundaries in members of Sect. *Cerris*, allowing occasional crosses. Indeed, our results demonstrate that *Q. trojana* and *Q. libani*, *Q. brantii* and *Q. macrolepis*, *Q. cerris* and *Q. suber*, and *Q. trojana* and *Q. suber* are interfertile and hybridize in the wild. Further investigations (e.g., with metagenomics approaches, fine-scale geographic samplings) are needed to distinguish between ancient hybridization with subsequent incomplete lineage sorting and retention of ancestral traits, to clarify the status of several other samples that may represent both phenomena based on shared 5S-IGS variants, the occurrence of unique sequence features, length of terminal branches and odd-placing in the phylogenetic reconstructions. Adequately addressing these issues would be of great relevance to identify relict populations and/or past contact/hybrid zones, to assess the hybridization ability of species growing in sympatry, and further define the evolutionary history of the *Cerris* oaks.

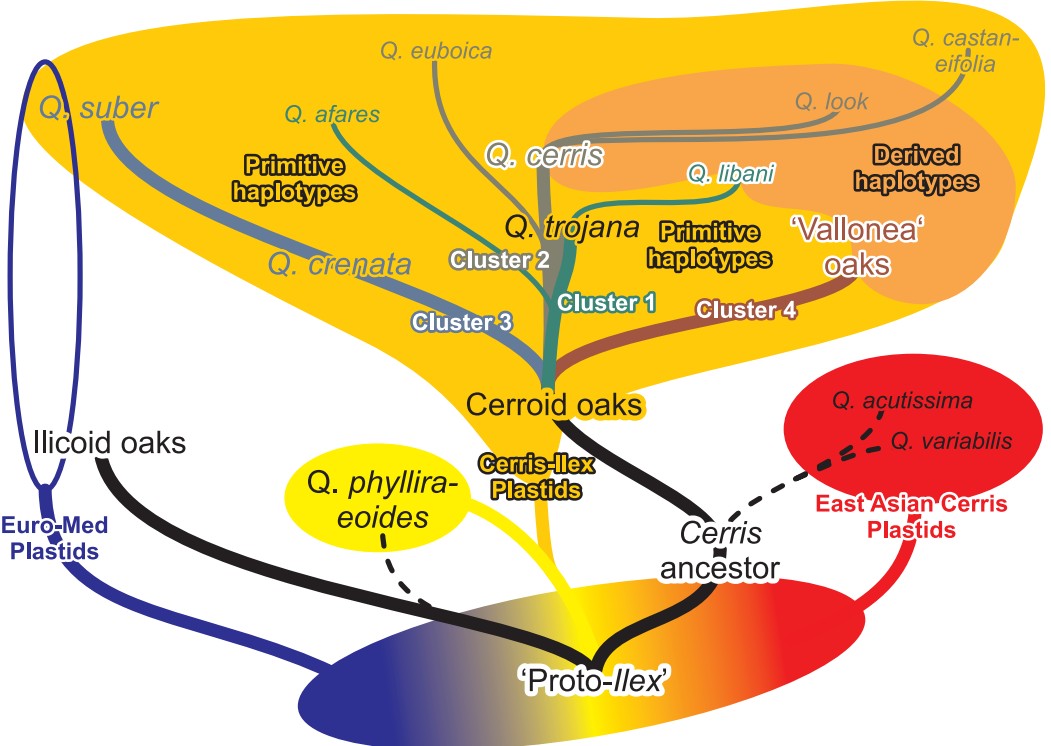

**Figure 6** **Mixed branching silhouette-tree doodle depicting the molecular differentiation and evolution in *Quercus* Section *Cerris*.** The evolutionary or genealogical lineages are indicated by branches (accordingly labelled and coloured), the fields represent shared or unique gene pools. The deep incongruence between plastid genealogies and nuclear-morphological phylogenetic lineages can only be explained by ancient reticulation and incomplete lineage sorting during the formation and isolation of the modern-day lineages following the break-up of the ancestral gene pool (tentatively labelled as 'proto-*Ilex*').

## Taxonomic framework of *Quercus* section *Cerris* in western Eurasia

From the clusters identified by the 5S-IGS network based on the average inter-individual clone (Fig. 4) and PBC-transformed distances (Fig. 5), four major groups can be identified representing distinct evolutionary lineages and used as a framework. Figure 6 shows a scheme, a cactus-type branching silhouette (*Podani, 2017*; *Morrison, 2018*), based on the 5S-IGS and *trnH-psbA* differentiation patterns, and with respect to the plastid tree provided in *Simeone et al. (2016)*.

The first, most western lineage includes *Q. suber* and *Q. crenata*. Sample cr05 likely represents a F1 hybrid with *Q. cerris*. The same holds for tj08, a *Q. trojana* x *Q. suber* hybrid. A second lineage, tentatively termed the 'occidentalis' lineage includes the widespread *Q. cerris* and the geographically restricted *Q. castaneifolia, Q. euboica* and *Q. look*. The third lineage collects the Eastern Mediterranean 'Vallonea' oaks *Q. brantii, Q. ithaburensis,* and *Q. macrolepis,* with three potential outliers: br02, br03 and it03 (possible recent or ancient hybrids). The last, least coherent group, the 'oriental' lineage, includes *Q. afares, Q. libani* and *Q. trojana*. Aside from supposed hybrids and introgressed individuals (see section above), these groups almost perfectly match previous taxonomic observations (sect.

*Heterobalanus* subsect. *Suber* + sect. *Cerris* subsect. *Cerris* and *Aegilops*; *Menitsky, 2005*; sect. *Suber* + sect. *Eucerris*, *Aegilops* and *Erythrobalanus*; *Schwarz, 1936–1939*), with the only exceptions of *Q. afares*, accomodated by both authors within the *Q. cerris-Q. castaneifolia* group, and *Q. crenata* and *Q. look* that were not included in previous monographs.

*Quercus trojana* is the only species scattered over two clusters (cluster 1 and 2, Figs. 4 and 5; see also *Denk & Grimm, 2010*), thus bridging between the 'oriental' and 'occidental' lineage. No geographic, haplotypic or subspecific relationships could explain this subdivision; it might therefore indicate the occurrence of two different, geographically overlapping but genetically isolated lineages within the species, possibly differentiated in the past (retained ancient polymorphism), especially considering the proximal positions of the sequences of samples tj24, 39 and 45 in the ML tree. Both *Q. trojana* (s.str.) sublineages occur in Italy and might correspond to the two main nuclear gene pools identified by *Carabeo et al. (2017)*. Indeed, more ecological and molecular data are required to interpret this finding biologically.

Overall, the complex genetic differentiation patterns can only be explained by longer ongoing free genetic exchange or more recent common origin of the 'oriental' and 'occidental' lineages than in case of the other two lineages within western Eurasian *Cerris*: the corkish oaks (*Q. crenata* + *Q. suber*) and the 'Vallonea' (or Aegilops) oaks (*Q. brantii*, *Q. ithaburensis*, and *Q. macrolepis*). *Quercus euboica* possibly originated by geographical isolation from a proto-*trojana* still close to the proto-*cerris* (Fig. 6). Interestingly, all the *Q. euboica* samples occurred in the *Q. cerris*- dominated 'crown' clade 4 (Fig. 3), hence, are part of the 'occidentalis' lineage. Likewise, the microspecies *Q. afares*, *Q. castaneifolia*, and *Q. look* as well as the eastern replacement of *Q. trojana*, *Q. libani*, were isolated from the master *cerris-trojana* genepool(s). It is impossible to provide absolute dates for the final isolation events. Comparison with intra- and inter-species 5S-IGS divergence in the other two main lineages of western Eurasian oaks ('ilicoid' oaks of sect. *Ilex*, 5S-IGS is species-diagnostic; 'roburoid' oaks of sections *Quercus* and *Ponticae*, 5S-IGS is largely undiagnostic; *Denk & Grimm, 2010*) indicates that the original split possibly predates diversification in the 'roburoid' oaks. The final establishment of species in western Eurasian members of Sect. *Cerris* is however as young as in the 'roburoid' oaks, may be ongoing (*Q. trojana*), and younger than the main split within the 'ilicoid' oaks, i.e., of *Q. ilex* from *Q. aucheri* (+*Q. coccifera*).

## Phylogeography of *Quercus* section *Cerris* in western Eurasia

The primitive (L1) and derived (L2) haplotype groups of 'Cerris-Ilex' lineage, and two haplotypes representing the 'Euro-Med' lineage characteristic for western populations of sect. *Ilex* (*Simeone et al., 2016*; *Vitelli et al., 2017*), describe the main evolutionary trajectories of the western Eurasian lineage of sect. *Cerris* and their contact with already established lineages of sect. *Ilex*. According to coalescent theory, the most frequent and widespread haplotypes, i.e., H1 and H2, are likely ancestral (*Posada & Crandall, 2001*; Fig. 1). The close relationship between haplotypes H1 and H2–found all across western Eurasia (Fig. 2B) and in all taxa except the south-eastern species group, *Q. brantii*, *Q. look*, and *Q. ithaburensis*—and *Q. phylliraeoides* (Fig. 5; cf. *Simeone et al., 2016*, Fig. 1), the only

species of section *Ilex* extending into Japan—points towards a north-eastern Asian origin of sect. *Cerris* and a westward migration of a large population into the Mediterranean region. The revised (see Introduction) fossil record of *Cerris* in (North-)East Asia (starting from early Oligocene) predates earliest records in western Eurasia (Oligocene/Miocene boundary) by ca. 10 Ma, thus rejecting the hypothesis that sect. *Cerris* evolved from the western stock of sect. *Ilex* populations with 'WAHEA' haplotypes (*Denk & Grimm, 2010*; *Simeone et al., 2016*). The effective population size of the early west-migrating *Cerris* must have been (very) large in contrast to their East Asian siblings. The East Asian species of sect. *Cerris* are more heterogenous (*Chen et al., 2012*; *Zhang et al., 2015*), and differ much more profoundly from *Q. phylliraeoides* (the Japanese *Ilex* oak), but also from the 'WAHEA' haplotypes of sect. *Ilex* and *Q. baroni* (Fig. 1), considered early diverged plastid lineages of subgenus *Cerris* ('Old World' or mid-latitude clade; the earliest diverged plastid lineage being the western Mediterraenan 'Euro-Med' type found in *Q. ilex*; Fig. 5; *Simeone et al., 2016*). In this context, assessment of the 5S-IGS diversity in (East) Asian members of sections *Cerris* and *Ilex* would be needed, to check whether the Asian counterparts are equally coherent or more diverse than their western Eurasian relatives.

Once established in the Mediterranean region (H1, H2; Fig. 2B), local bottlenecks may have contributed to increased genetic drift in the plastome in the eastern part of the range. A likely trigger are the complex orogenies shaping modern-day Turkey and the Levant, areas with an increased haplotype diversity including the most derived 'Cerris-Ilex' haplotypes (Figs. 1–2). This, and the general west-east differentiation pattern (see also Figs. 4 and 5), parallels the situation in sect. *Ilex*, *Q. coccifera* in particular (*Vitelli et al., 2017*). A notable difference to sect. *Ilex* is the lack of plastid structuring (and diversity) in the central and western Mediterranean region, indicating a rather recent, singular colonization by the master population, clearly not affected by Oligocene micro-plate tectonics as suggested for *Q. suber* by *Magri et al. (2007)*.

The derived L2 'Cerris-Ilex' haplogroup (H5–H10) starts in Anatolia and extends further east (Iran) and south (Levant). In addition to isolation during range establishment, specialization to drier climates (e.g., summer-dry Mediterranean climates: *Csa, Csb, Dsb*) can be considered as trigger for increased genetic drift, possibly linked to speciation. The Aegilops oaks, *Q. brantii*, *Q. ithaburensis*, and *Q. macrolepis*, a well-circumscribed group based on 5S-IGS differentiation (Figs. 4 and 5) and morphology, are unique by showing only derived 'Cerris-Ilex' haplotypes.

A remarkable exception are two Italian *Q. cerris* individuals showing the derived haplotypes H9/H10, which occur in locations more than 2,000 km apart from other individuals of this Levantine haplotype sublineage. H9/H10 derive from types found in Anatolia and eastwards (Figs. 1 and 2). Long-distance seed dispersal is highly unlikely. The main animal vector for propagation of oaks are the jaybirds, which are sedentary birds, with a short evasion range (<50 km; *Pesendorfer et al., 2016*). Man-mediated dispersal (in historic times) could be a likely explanation, although we note that haplotypes shared by disjunct central Mediterranean and the Anatolian regions were also found in *Q. ilex*, and possibly reflect the remnants of a pre-Quaternary continuous range.

In this context, the genetic diversity detected in the Italian *Q. trojana* populations (both at the nuclear and at the plastid level) and the very limited, amphi-Adriatic distribution of haplotype H2 in *Q. macrolepis* (Italy, Albania; Fig. 2, File S1) likely confirm that these oaks are native in Italy. Similar close intra-specific phylogeographic relationships have been detected in other plant species on both sides of the Adriatic Sea (*Musacchio et al., 2006*; *Hilpold et al., 2014*), including oaks (*Lumaret et al., 2002*; *Fineschi et al., 2002*; *López de Heredia et al., 2007*; *Bagnoli et al., 2016*). In this case also, the Apulian populations of *Q. trojana* and *Q. macrolepis* can be interpreted as the remnants of a once continuous ancestral range (*Simeone et al., 2016*), or witness a colonization wave that was likely favoured by land connections between the Balkans and southeastern Italy during the Messinian salinity crisis and (or) the Pleistocene glaciations (*Nieto Feliner, 2014*).

## CONCLUSION

The present study is the first to include all putative species of *Quercus* sect. *Cerris* in western Eurasia. Our investigation is based on a dense intra-specific and geographic sampling and makes use of DNA sequence variation of the two most divergent nuclear and plastid regions known for oaks. Although based on just two markers, the obtained results confirm and emend species relationships and the genetic coherence of taxa. An updated subsectional classification of the western Eurasian *Cerris* oak species is proposed, with the identification of four major lineages, corresponding to subsectional groups that would need to be formalized. Some intraspecific taxa are recognized as distinct species (i.e., *Q. macrolepis* and *Q. euboica*) and the systematic relationships of *Q. look* are clarified. Although we observed the occurrence of occasional F1 hybrids, possible intrograded individuals and several potential outlier individuals across the studied range, we could not confirm the hybrid origin of *Q. afares* and *Q. crenata*. The fossil record corroborates major inferences about the origin and diversification of the section.

Characterizing nuclear and plastid differentiation across all species, including numerous individuals and the entire range, can only be the first step. Figure 6 summarizes our results, but also highlights phenomena deserving further investigation. Primarily, 5S-IGS data need to be compiled for (East) Asian members of sections *Cerris* and *Ilex*. A future focus should be on all Hindukush to western Himalayan species and the Japanese *Q. phylliraeoides,* the north-easternmost member of sect. *Ilex*, which has plastid signatures very similar to the western Eurasian members of sect. *Cerris* but not to the geographically closer East Asian species of sect. *Cerris*. The entire fossil record of sections *Cerris* and *Ilex* should then be recruited to infer age estimates, following the recent example of the genus *Fagus* (*Renner et al., 2016*). Another open question is where to root the nuclear tree (the polytomy in Fig. 6): our incomprehensive outgroup places the root within the *crenata-suber* portion of the 5S-IGS ML tree, which would mean that the 'corkish' oaks represent the first diverging lineage. This rooting hypothesis does not fit well with the structure of the PBC network and is in conflict with plastid and fossil evidence favouring a north-eastern origin of the section. A stepwise East to West invasion of sect. *Cerris* into the Mediterranean region is also supported by higher species and plastid diversity in the East Mediterranean. It

is possible that the westernmost ancestral populations of the cerroid oaks, carrying the common haplotype, went through a relatively recent bottleneck resulting in unique and distinct 5S-IGS variants. These distinct 5S-IGS variants would then be attracted to any possible (distant) outgroup when inferring a tree (ingroup-outgroup branching artefact; cf. position of outgroups in Fig. 4). Finally, *Q. cerris* should be investigated in detail across its entire range using a combination of morphometric and high-resolution genetic analysis to elucidate its relationships with sympatric species of sect. *Cerris* and the isolated endemisms. This will allow us to test whether *Q. cerris* is a primal genetic and ecological resource of the section in western Eurasia and carrier of ancestral signals.

### Funding
This project was funded by a Swedish Research Council (VR) grant to Thomas Denk. Sequencing was made possible using funds from a VR 'ForsAss' grant (no. 2008–3726) to Guido W. Grimm. Guido W. Grimm has been funded by the Austrian Science Fund FWF 'Lise-Meitner mobility' grant (project no. M1751-B16). The funders had no role in study design, data collection and analysis, decision to publish, or preparation of the manuscript.

### Grant Disclosures
The following grant information was disclosed by the authors:
Swedish Research Council (VR).
VR 'ForsAss': 2008–3726.
Austrian Science Fund FWF 'Lise-Meitner mobility': M1751-B16.

### Competing Interests
The authors declare there are no competing interests.

### Author Contributions
- Marco Cosimo Simeone conceived and designed the experiments, performed the experiments, analyzed the data, contributed reagents/materials/analysis tools, prepared figures and/or tables, authored or reviewed drafts of the paper, approved the final draft.
- Simone Cardoni performed the experiments, prepared figures and/or tables, approved the final draft.
- Roberta Piredda analyzed the data, prepared figures and/or tables, authored or reviewed drafts of the paper, approved the final draft.
- Francesca Imperatori performed the experiments, contributed reagents/materials/analysis tools, approved the final draft.
- Michael Avishai and Thomas Denk conceived and designed the experiments, contributed reagents/materials/analysis tools, authored or reviewed drafts of the paper, approved the final draft.
- Guido W. Grimm conceived and designed the experiments, analyzed the data, contributed reagents/materials/analysis tools, prepared figures and/or tables, authored or reviewed drafts of the paper, approved the final draft.

## DNA Deposition

The following information was supplied regarding the deposition of DNA sequences:

All sequence data generated as part of this study are available on GenBank: accession numbers LT963519–LT963530 and LT971421–LT972071.

## Data Availability

Primary data and analyses are available as Supplemental Files.

## Supplemental Information

Supplemental information for this article can be found online at http://dx.doi.org/10.7717/peerj.5793#supplemental-information.

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
