# Peer review of "Comparative systematics and phylogeography of Quercus Section Cerris in western Eurasia: inferences from plastid and nuclear DNA variation"

_PeerJ, doi:10.7717/peerj.5793_

## Round 0.1 · original submission · Minor Revisions

Dear Marco Cosimo,

Both reviewers agree on the general value of your contribution, but also both make several minor suggestions that could contribute to improve it.
Please, follow their suggestions and re-submit your manuscript.

Thanks!

Sincerely,

Marcial.

Reviewer 1 ·

Basic reporting

The authors present an impressive amount of work on a challenging but very interesting question about the relationships among Sect. Cerris oaks of western Eurasia. Presenting the many detailed and variable results among markers and species presents difficulties for the reader in fully appreciating where the data tell the story and where the authors interpretation of those data are driving the story. The length and readability of the manuscript presented the largest challenges to me. The questions are very interesting, the analyses are all justifiable, and the figures are extremely helpful in clarifying the interpretation of the results. The colors are nice and consistent but If the publication allows, individual photos of each species (close up of 1 leaf) may be helpful in keeping track of them as well.

Experimental design

Starting at line 130, I’m not sure if the challenges presented are specific to your system/seciton or the entire genus where a lot of that type of work has already been done. It also seems like a list of general challenges that seemed to slow the momentum of your introduction section. Maybe these methodological details could be broached in the Methods section or in the Discussion? Just a thought as I found lines 130-147 to be a little less clear than the rest of the introduction.

Generally, it seems you should keep the focus on Sect. Cerris and focus less on the broader patterns in the the genus Quercus unless there are similar phytogeographic patterns that might serve to highlight expectations or contrasts with this study. This might help focus the manuscript a bit as well as allow for the elimination of some of the more tangential information about broader Quercus

Validity of the findings

I understand the challenges in finding appropriate molecular markers but it should be emphasized that most studies rely on more than 1 nuclear and 1 plastid sequence region to infer relationships to the degree that authors set out to in this study. An upfront explanations of potential caveats about inferences made from so few markers should explain the degree of uncertainty in drawing conclusions from only these markers.

The Discussion is very long but does a good job of identifying all the additional hypotheses that are raised by the results presented here. The use of subheadings is helpful to the reader but I encourage the authors to try to find sections that could be more concise by reducing the amount of speculative interpretation.

Overall, this manuscript seems like an important first step in many future more detailed studies of these oaks.

Additional comments

INTRODUCTION
86 "Section Cerris (Cerris oaks) currently comprises 12 or 13 species (Table 1) occurring from the"
In the abstract you say 15 species, I think I understand why this difference but it seemed odd reading these different numbers so close together without clarifying information.

64 "Oaks (Quercus L.) are a worst case and, at the same time, an ideal model for comparative
65 phylogeographic studies."
Clarify this statement by saying immediately following it with "because they . . . " or start the next sentence with “They are ideal because . . . but challenging because . . . “

89 "with different precipitation regimes (Cs, Cf, Cw, Ds, Df, Dw; Kottek et al., 2006;"
Readers may not be familiar with codes for precipitation regimes - could you summarize the range with words: arid - mesic?

METHODS
I am unfamiliar with the Goker and Grimm (2008) method the authors used so I cannot comment on the appropriateness of each step but I appreciate the transparency describing the steps taken.

I didn’t follow the rational for reversing the inverted sequences?

RESULTS

I may have missed the explanation for why or how the root of the tree was chosen?

294 "sections in the Aegean and the western Mediterranean regions. As shown in Fig. 2a-b, plastid

295 diversity is largely decoupled from species identity, and related to geography; the least derived

296 haplotypes within the ‘Cerris-Ilex’ lineage (H1 and H2) occur across the whole distribution

297 range of the investigated group, except the Levant and the western Mediterranean (H2). All other

298 haplotypes are more circumscribed, concentrated in Anatolia (H3–H5, H7–H8), the Levant (H6,

299 H9–H10; the latter two showing single occurrences in Italy), and Iberian Peninsula + Morocco

300 (H11-H12)."
This is a cool result and it seems to be a major outcome of the study! It might help the reader to hear more about nuclear-plastid incongruence in the Introduction to set some expectations and may provide an opportunity to explain

Was there any thought of trying to better characterize the putative hybrids using methods like those summarized in the Twyford and Ennos 2012 Heredity paper “Next generation hybridization and introgression” (Heredity volume 108, pages 179–189): "The genomic contribution of the parental lineages in each hybrid individual can then be estimated (the ‘hybrid index’ or ‘admixture proportion’; maximum likelihood implementation in HINDEX; Buerkle, 2005) as well as the hybrid class (for example, F1, F1 backcross, model-based Bayesian implementation in NEWHYBRIDS; Anderson and Thompson, 2002). Moreover, where detailed genomic data are available, genomic clines of introgressed alleles can be identified (using INTROGRESS; Gompert and Buerkle, 2010)."

Although most of those methods do require broad marker coverage throughout the genome. Or maybe what you have in Fig. 4 does something similar to this?

DISCUSSION

456 "Quercus cerris has the largest range and broadest climatic envelope (from perhumid Cfa, Cfb via"
This section of the discussion that brought in the climatic and ecological data was interesting and the reader could be more prepared for these results with clearer expectations about the role these factors could potentially play in shaping genetic diversity.

506 "This oak grows isolated from

507 Q. trojana on the Greek island of Euboea and differs morphologically by its coriaceous leaf

508 texture and the conspicuous white tomentum of the abaxial leaf surface that is made up of stellate

509 trichomes (T. Denk, pers. observ.) In addition, Q. euboica is characterized by special edaphic"
Can all these morphological and ecological characteristics of the species/populations be summarized in some kind of clustering approach that might allow comparisons between genetic variables and morpho/ecological variables?

515 "grouped (mostly) in a specific sublade (Fig. 3, File S4). However, more (morpho-ecological) "
should be “subclade” I think

Reviewer 2 ·

Basic reporting

see below

Experimental design

see below

Validity of the findings

see below

Additional comments

Simeone et al

This paper nicely presents the genomic landscape of section Cerris in western Eurasia based on impressive sampling of plants, two genomes, and more or less standard sets of analyses. The substantial break between Q. acustissima and Q. variabilis at the plastid level is interesting and a bit underplayed in the paper. It would have been nice to have that end of the diversity within Cerris included in the the 5S-IGS study. If those sequences are available, I encourage their inclusion. The introduction could use some structure that would help the reader understand the research objectives associated with Cerris, but also to clearly lay out the possible zones of contact with Ilex species. As these are potentially known, the added sampling here is useful in uncovering other areas of historical contact. The previous study of Ilex is important, but it seems that the goals for the study of Cerris are somewhat obscured in the Introduction. The main point is unpack the Introduction a bit, and to let the questions related to Cerris oaks drive the study.

Abstract:

Line 31 – “species coevolution” – not really what is intended, so change to -- reflected phylogeographic patterns originating from likely interspecific cytoplasmic gene flow within…
Line 33 – “siblings” -- change to descendent species, related species or sister clade or whatever might work better.

Introduction:

Line 64 -- “worst case” – don’t agree in this context, but if you want to convey that idea, it should be justified. In the past, the “worst case” qualifier has been applied to oaks based on failure to meet the criteria for particular species concepts, but I don’t think this section was going there.

Line 90 – you might use leaf persistence or leaf lifespan, eg, Leaf lifespan is quite variable, ranging from…

Lines 127-129 – Consider turning these couple of sentences into a general objective statement for the study.

Lines 141 – 35S rDNA? Is 35S correct?

Lines 146 and on – this is an important paragraph but it seems to focus mostly on Ilex phylogeography. Consider using this as opportunity to better set-up the study of Cerris. For example, what are the main issues? Will related species show regional patterns of distribution? Where are the main contact zones with other Cerris and Ilex species? Is the signature of that contact detectable in both data sets? Perhaps add something about the hybrid origin of Q. afares (suber x canariensis), which is of broad interest as the only case of a suspected naturally occurring intersectional hybrid species. It’s good to put that concept to rest. These are just off the top of the head, so there could be more.

Line 150 – phylogenetic quality? -- each main plastid group has a distinct geographic distribution that encompasses parts of the taxonomic breadth of sect Ilex and Cerris…not sure if that gets to what you mean?

Line 152 – “WAHEA” -- define this acronym -- perhaps by describing the main component bio regions that contain plastid lineages etc.

Materials and methods:

Consider clarifying the amount of new data that is being presented relative to the proportion of previously published data. This would provide a better understanding of how the new sampling and data fill the holes needed to address Cerris and interactions with Ilex. I see the details in the Results, so perhaps consider this an opportunity to develop a sampling design statement.

For example, it seems that comparing east Asian diversity of Cerris is challenging at the plastid level, eg, divergent, and no shared haplotypes, and unfortunately not sampled for the 5S-IGS study. That seems like a weakness in the sampling design.

Results:

Figure 2/3. Nice. Consider crossing referencing the biogeo areas (occidental, Vallonea, etc.) referred to in Figure 5 within the inset for the 4 groups in panel C. Perhaps do something similar with haplotypes if possible.

Discussion:

Mostly good and informative. It could be shortened by transferring certain descriptive aspects to the Results. I encourage the authors to consider streamlining this section if at all possible.

Line 437 – “confirmed their status as most variable” -- I get that but is this source of variation the best choice? I’m not so sure given the most of inheritance, need to clone, etc. I guess I’m thinking that 1000s of single copy genes would be preferred because of the potential to look at allele trees from many loci using a range of species tree gene tree reconciliation methods. I also wonder whether analyses designed to reveal levels of admixture, like STRUCTURE, would be appropriate. The approach used here to identify hybrids appears to be somewhat ad hoc and visual, based on shared presence of gene copies that share network edges etc.

Line 442 – “Combined data” -- this sounds like an analytical process – Perhaps try something like, Insights gained from the combination of data sources were concordant. It’s important to recognize that these insights are not based on a formal correlation analysis, but It would be interesting to consider a spatial auto-correlation approach to the two sets.

Line 608-618 – the writing in this section could use some work.

“derivedness” maybe try divergently derived in another sentence structure?

“nor” – and there was no obvious phylogenetic linkage between the Q. afares and the Q. crenata-suber lineage.

619-630 – this is a great point that will no doubt require looking at 500 or more loci and examining parental contributions at the allele tree level. And now, with multiple oak genomes available, the possibilities for fine scale analyses using gene tree species tree reconciliation methods is possible. The ability to discriminate current vs historical patterns of introgression is another way to possibly frame the interaction zones described throughout the paper.

---

## Round 0.2 · accepted · Accept

Dear Marco Cosimo,

Congratulations!

You manuscript is accepted for publication in PeerJ.

Please, make sure to edit all your tables. Replace "," with ".".
Sincerely,

Marcial.

#